# Fat body glycolysis defects inhibit mTOR and promote distant muscle disorganization through TNF-α/egr and ImpL2 signaling in *Drosophila* larvae

Miriam Rodríguez-Vázquez [ID][1], Jennifer Falconi [ID][1], Lisa Heron-Milhavet [ID][1], Patrice Lassus [ID][2], Charles Géminard [ID][1,3] & Alexandre Djiane [ID][1,3][✉]

## Abstract

**The fat body in *Drosophila* larvae functions as a reserve tissue and participates in the regulation of organismal growth and homeostasis through its endocrine activity. To better understand its role in growth coordination, we induced fat body atrophy by knocking down several key enzymes of the glycolytic pathway in adipose cells. Our results show that impairing the last steps of glycolysis leads to a drastic drop in adipose cell size and lipid droplet content, and downregulation of the mTOR pathway and REPTOR transcriptional activity. Strikingly, fat body atrophy results in the distant disorganization of body wall muscles and the release of muscle-specific proteins in the hemolymph. Furthermore, we showed that REPTOR activity is required for fat body atrophy downstream of glycolysis inhibition, and that the effect of fat body atrophy on muscles depends on the production of TNF-α/egr and of the insulin pathway inhibitor ImpL2.**

**Key words** Glycolysis; Inter-Organ Communication; Adipose Tissue; Muscle Wasting; *Drosophila*
**Subject Categories** Metabolism; Signal Transduction

## Introduction

In animals, complex exchanges between the different organs are central to their physiology and to their harmonious growth. In particular, coping with intermittent nutritional resources or developmentally evolving metabolic needs during growth and metamorphosis, reserve organs allow the storage and subsequent release of energy and nutrients when needed.

The fat body of *Drosophila*, or adipose tissue, represents the main organ for storing energy and nutrients in the form of sugars (glycogen), proteins, and lipids. These reserves are stored or released in response to external (dietary) and internal (hormones and cytokines) stimuli. In case of excess energy, a high Insulin pathway in adipocytes promotes the storage of circulating carbohydrates (glucose or trehalose), as glycogen or lipids. Lipids stored in fat body cells come either from dietary lipids absorbed by the gut and transported to the adipocytes, or are synthesized within adipocytes through neo-lipogenesis. Core metabolism links glycolysis, the mitochondrial tricarboxylic acid (TCA) cycle, and neo-lipogenesis: pyruvate, the final product of glycolysis, enters the mitochondrion and the TCA cycle to produce citrate that will serve as a precursor to produce fatty acids which are then incorporated into lipids. Alterations of the TCA cycle, as observed in Seipin mutants (Ding et al, 2018) or in more recent studies tackling the function of mitochondrial activity by analyzing *TFAM* knock-down (Sriskanthadevan-Pirahas et al, 2022), show that lowering mitochondrial and TCA cycle activities result in lipid droplets and lipid storage shrinkage.

During larval stages, nutrient sensing and mTOR pathway regulation in adipocytes control organismal growth (Colombani et al, 2003; Géminard et al, 2009). The evolutionary, highly conserved mTOR pathway represents a key cellular metabolism integrator and regulator (Bjedov and Rallis, 2020). The mTor kinase, together with co-factors including raptor, forms the TORC1 complex which then phosphorylates a wide array of substrates leading to metabolic adaptation (here increased metabolism in high amino-acids availability; (Bjedov and Rallis, 2020)). More specifically, TORC1 phosphorylates and activates Ribosomal protein S6 kinase (S6k), thus promoting ribosomal activity (Saitoh et al, 2002). TORC1 also phosphorylates several transcription factors, but in *Drosophila* most of the transcriptional program downstream of the mTOR pathway is controlled by REPTOR. TORC1 phosphorylates and inactivates the transcriptional co-activator REPTOR, preventing it from entering the nucleus and to associate with its transcriptional partner REPTOR-BP (Tiebe et al, 2015).

Besides its reserve function, the fat body of *Drosophila* constitutes thus a central endocrine organ. In response to changes in their metabolism, fat body cells send different adipokines which will impact many organs and control remotely general organismal metabolism and growth (Ahmad et al, 2020; Meschi and Delanoue, 2021). In particular, many adipokines control the release of insulin (insulin-like peptides dilp2 and dilp5) from the insulin-producing cells (IPCs) located in the larval brain (Ahmad et al, 2020; Meschi

[1]IRCM, Univ Montpellier, Inserm, ICM, Montpellier, France. [2]IRCM, Univ Montpellier, Inserm, ICM, CNRS, Montpellier, France. [3]These authors contributed equally: Charles Géminard, Alexandre Djiane. [✉]E-mail: alexandre.djiane@inserm.fr

and Delanoue, 2021). In response to varying nutritional inputs and changes in mTOR signaling, adipocytes thus produce and secrete factors either promoting (stunted/sun; Growth Blocking Peptides 1&2/Gbp1&2; upd2, and CCHamide-2/CCHa-2) or preventing (TNF-α/egr) dilps production and secretion from IPCs (Agrawal et al, 2016; Delanoue et al, 2016; Ingaramo et al, 2020; Koyama and Mirth, 2016; Rajan and Perrimon, 2012; Sano et al, 2015).

Interestingly, the control of Insulin pathway activity is not restricted to the regulation of dilps secretion, but is also achieved by extracellular proteins such as ImpL2, dALS, and Sdr (Alic et al, 2011; Arquier et al, 2008; Bader et al, 2013; Honegger et al, 2008; Okamoto et al, 2013). These proteins can trap dilps in the hemolymph and are thought to act mainly as inhibitors. Upon reception by target cells, dilps activate the Insulin pathway to favor anabolic processes and the implementation of growth or repair programs. In Drosophila muscle cells, Insulin signaling plays a role equivalent to IGF-1 signaling in mammals (Yoshida and Delafontaine, 2020) and prevents, at least in adults, muscle ageing and the formation of detrimental protein aggregates (Demontis and Perrimon, 2010; Wessells et al, 2004).

In this study we analyzed the role of glycolysis knock-down on fat body biology and on the development of Drosophila larvae. Glycolysis shutdown in adipocytes triggered a dramatic lipid reserve shrinkage and fat body atrophy. Fat body atrophy was also accompanied by a delay in pupariation, and in distant body wall muscle disorganization, suggesting the remote control by fat body cells of muscle cell integrity. Molecularly we further show that glycolysis shut down in adipocytes resulted in low mTOR and REPTOR activities, and triggered the release of the TNF-α/egr adipokine, and the production of the dilps inhibitor ImpL2. We propose that the combined actions of increased TNF-α/egr and low Insulin are responsible, at least in part, for the muscle wasting triggered by distant fat body atrophy.

# Results

## Impaired glycolysis in adipose tissue leads to adipose tissue atrophy

To investigate the role of adipose tissue metabolism on the general development of the fly, we targeted glycolysis. Using lpp-Gal4 to drive inducible RNAi in fat body cells, we probed the function of different glycolysis enzymes (Fig. 1A) on the size and morphology of adipose tissue cells. Knocking down the expression of the early enzymes of glycolysis did not have any significant effects, probably due to insufficient knock-down, potential redundancy in enzyme function, or existing alternative pathways such as the pentose shunt (Fig. 1A). However, knocking down the enzymes controlling the later steps of glycolysis, such Phosphoglyceromutase (Pglym78) or Enolase (Eno) resulted in smaller and atrophic fat body. First, Pglym78-RNAi or Eno-RNAi treated cells were significantly smaller than cells expressing a control RNAi (against the gene white). Average cell size were 55 and 75% smaller than controls, respectively (Fig. 1B,C). The knock-down of Phosphoglycerate kinase (Pgk), the glycolytic enzyme just above Pglym78, had a weaker effect under these conditions (Fig. 1B,C). Importantly, the knock-down efficiencies of the different RNAi used were very similar and reached more than 80% for Pgk, Pglym78, and Eno,

suggesting that the differences observed between Pgk and Pglym78 were not caused by differences in RNA knock-down (Fig. EV1A). Then, both Pglym78-RNAi and Eno-RNAi cells had fewer and smaller lipid droplets than controls as monitored by BODIPY stain (Fig. 1B,D), as well as smaller amounts of triglycerides (TAG, Figs. 1E and EV1D). Similar results obtained knocking down two different steps of the glycolysis using two different RNAi constructs strongly argue that the effects observed were not unspecific effects of the tools used. Furthermore, similar results were obtained with ppl-Gal4, a second adipose-specific driver showing that upon glycolysis shut down, fat body cells undergo strong atrophy with smaller lipid reserves (Fig. EV1B,C). While lipid reserves were depleted, carbohydrate reserves were not affected in lpp-Gal4> Pglym78-RNAi animals, as shown by constant glycogen reserves in fat body cells and similar circulating sugar levels (trehalose; Fig. EV1E,F).

The end product of glycolysis, pyruvate represents a central metabolite that can be converted to Ac-CoA and then enters the TCA cycle in mitochondria to produce energy and other metabolites. Alternatively, Ac-CoA can be generated through the β-oxidation of lipids and fatty acids. We thus monitored the relative abundance of pyruvate and ATP, in the affected fat body cells. Even though glycolysis blockade led to severe fat body atrophy, fat body cells from lpp-Gal4> Pglym78-RNAi animals had increased levels of ATP and pyruvate (Fig. EV1G,H). The increased ATP levels, together with decreased triglycerides and lipid droplets (Fig. 1D,E), could indicate that Pglym78-RNAi cells, resorted to alternative metabolic pathways, and might have privileged fatty acid β-oxidation as a source of energy. Accordingly, the increase in pyruvate concentration might be explained by this adaptation, where cells stopped using pyruvate which thus accumulated, even though produced at a lower rate due to glycolysis blockade. It is noteworthy that the measurements used reflect steady-state levels, which might not be correlated with the actual usage of the different metabolites.

In order to support the link between glycolysis alteration and lipid droplet atrophy in our model, we then tested whether supplementing animals with pyruvate would bypass the effects of glycolysis knock down, as was shown previously (Ding et al, 2018). Growing lpp-Gal4> Pglym78-RNAi larvae on a feeding media supplemented with 1% pyruvate restored lipid droplet content, albeit not completely (Fig. 1B–D). Monitoring pyruvate levels inside cells showed that Pglym78-RNAi fat body cells from animals grown on pyruvate-supplemented food had less pyruvate than those from animals grown on standard food (Fig. EV1H), consistent with Pglym78-RNAi cells reverting to pyruvate consumption to generate Ac-CoA and fuel the TCA cycle, thus sparing lipids.

These results support the model in which lipid depletion and lipid droplets atrophy upon lower glycolysis knock-down are a consequence of reduced pyruvate consumption, resulting in metabolic rewiring and increased fatty acid β-oxidation. Low pyruvate usage, would also lead to lower de novo lipogenesis, consistent with earlier reports showing that Seipin/SERCA knock-down affected lipid reserves in Drosophila, in part through impaired glycolysis (Ding et al, 2018). However, we cannot exclude that the supplied pyruvate could affect other organs important for fatty acids and lipids synthesis, such as enterocytes or oenocytes, ultimately resulting in the sparing of lipid reserves in the larval fat body.

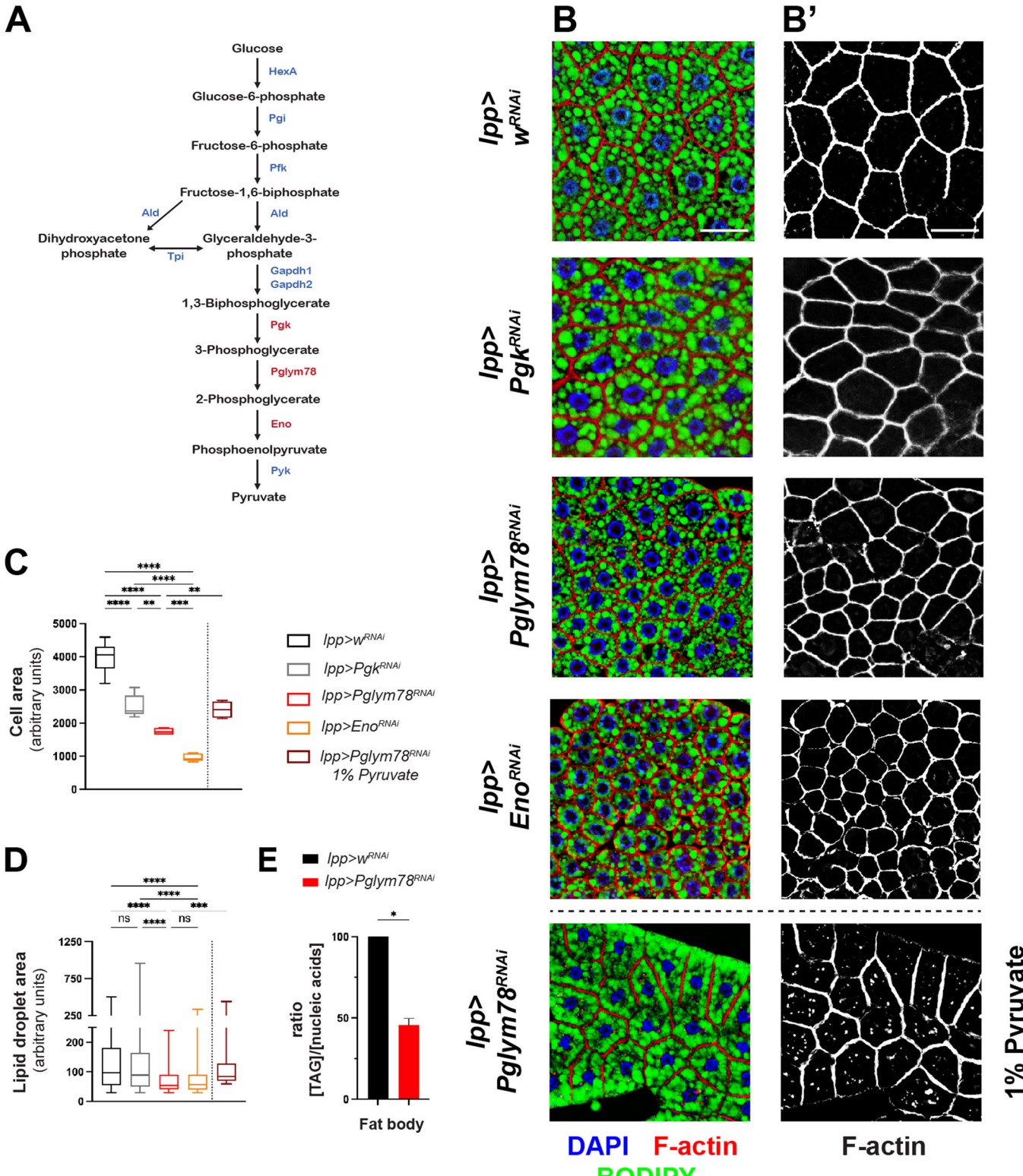

## Adipose tissue atrophy promotes body wall muscle atrophy and disorganization

We then questioned what were the effects on larval growth of the adipose tissue atrophy after glycolysis shut down. *Lpp-Gal4>*

*Pglym78-RNAi* larvae showed a developmental delay of around 48 h compared to controls, and larvae entered pupariation at day 7 rather than day 5 after egg laying (ael; Figs. 2A and EV2A). However, we did not observe dramatic changes in larval brain size or growth at 5 days ael (compared to white RNAi controls),

**Figure 1. Lower glycolysis enzymes knock-down in adipose tissue results in fat body atrophy.**

(A) Schematic of the glycolysis pathway with the key enzymes in *Drosophila*. (B) Fat body staining of the indicated genotypes showing nuclei (DAPI, blue), cell cortex (F-actin, red, white in B'), and lipid droplets (BODIPY, green); white bar: 50 μm. (C, D) Quantification of average cell size (C) and lipid droplet area per cell (D) from images shown in (B) and represented as box plots where the whiskers represent the maxima and minima experimental points, the boxes represent the 25 and 75% percentiles, and where the center line is the median. Biological replicates $n = 6$–10. One-way ANOVA (Kruskal–Wallis) statistical test. Tests in (C): ****$p < 0.0001$, ***$p = 0.0006$, ** 0.01, ns not significant. Tests in (D): ****$p < 0.0001$, ***$p = 0.001$, ns not significant. (E) Quantification of triglyceride (TAG) content in dissected fat bodies normalized to total nucleic acid content. Biological replicates $n = 4$. Error bars show the standard error of the mean (sem). *T*-test *$p = 0.0286$. Source data are available online for this figure.

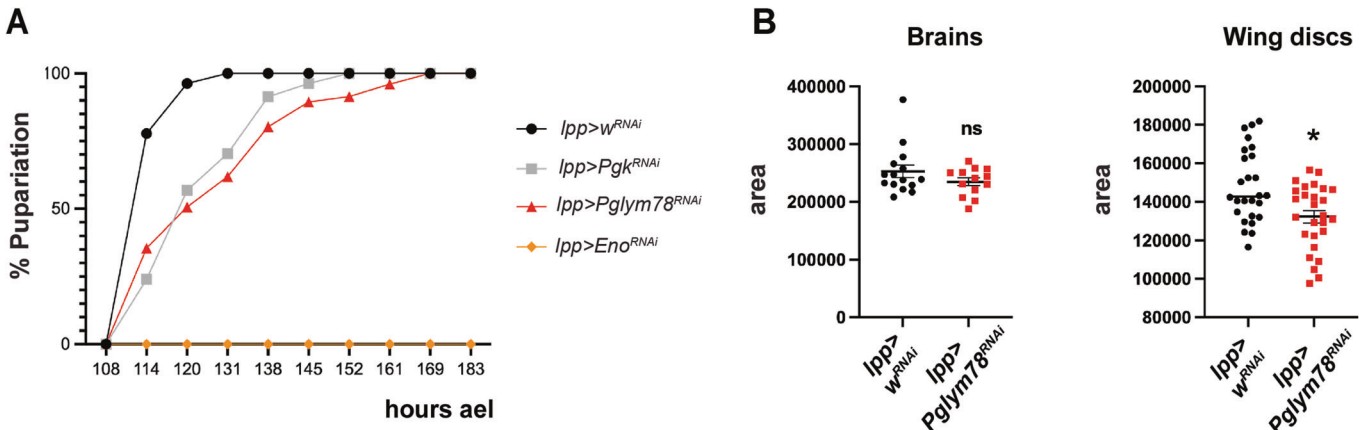

**Figure 2. Developmental delay after glycolysis enzymes knockdown in the fat body.**

(A) Pupariation curve of the indicated genotypes as cumulative percentage of pupae with time in hours after egg laying (ael). (B) Size of brains and wing disks expressed in arbitrary units (pixels) in *Pglym78-RNAi* animals compared to *w-RNAi* controls at 5 days ael. For brains, $n = 15$ and 13, for disks, $n = 26$ and 27. Error bars show the standard error of the mean (sem). Mann–Whitney test, *$p = 0.0175$, ns not significant. Source data are available online for this figure.

consistent with the existence of brain-protective mechanisms under nutritional or energetic stress (Cheng et al, 2011). There was, however, a small reduction in the size of wing imaginal disks (Fig. 2B). Similar pupariation delay was also observed in *Pgk-RNAi* animals (Fig. 2A). Interestingly, in these animals, the fat atrophy was much more subtle, and we did not detect significant lipid droplets shrinkage (Fig. 1B–D), suggesting that either developmental delay and fat atrophy might be two independent processes, or that even subtle effects on lipid content might be sufficient to delay pupariation. Furthermore, when we compared similarly delayed *Pgk-RNAi* and *Pglym78-RNAi* animals at 6 days ael, we did not detect any difference in the size of brains and wing disks, suggesting that fat atrophy does not directly control brain and disks growth in this model (Fig. EV2B). Animals depleted for *Eno* had the strongest developmental delay and never entered pupariation (Fig. 2A).

But while we did not detect alterations in the growth of the brain and disks of *Pglym78* depleted animals, body wall muscles were affected. Whole body wall muscle preparation, showed that upon *Pglym78* or *Eno* knock-down in the larval fat body, several muscles showed abnormal morphology and appeared shorter and condensed, as evidenced by bright F-actin staining (Figs. 3A,C and EV3A for phenotype range). Furthermore, the archetypical arrangements of muscles under the cuticle, either longitudinally or obliquely within segments, was perturbed in the affected animals. Focusing more specifically on the large VL3 and VL4 muscles located just laterally to the ventral midline in each larval segment, F-actin

staining was weaker than in controls, suggesting a reduction in muscle fiber density (Figs. 3A and EV3A). These different defects were not a consequence of poor handling of the larvae and dissection artifacts, since we could also document condensed muscle material (bright dots) using a transgenic line carrying a live GFP muscle marker (Zasp66::GFP protein trap fusion; Fig. 3B). These Zasp66::GFP features were reminiscent of the condensed severed muscles and of the dimmer fibers observed by F-actin staining on dissected and fixed cuticles.

Given the disorganization of body wall muscles, we next wondered whether muscular proteins could be found in the hemolymph. Collecting hemolymph from animals by gentle bleeding (see Materials and Methods), we could detect by western blot analyses α-Actinin, and Zasp66::GFP in the blood of *lpp-Gal4> Pglym78-RNAi* larvae. These proteins are highly enriched in muscles, and were not found in the blood of control animals, (*lpp-Gal4 > w-RNAi* or *Pgk-RNAi* for monitoring at 5 days or 6 days ael, respectively), confirming that following fat body atrophy, muscle proteins were released in the circulation (Fig. 3D). Importantly, α-Actinin was not detectable by western blot analyses on whole protein extracts from dissected fat body tissues (Fig. EV3B), further supporting the muscle origin of the muscular proteins detected in the hemolymph. It is noteworthy that the proteins detected were not due to contamination by circulating cells (hemocytes) or large cellular debris, since they were still detected in the supernatant after high-speed centrifugation of the collected hemolymph (Fig. EV3C). Strikingly, pyruvate-supplemented food

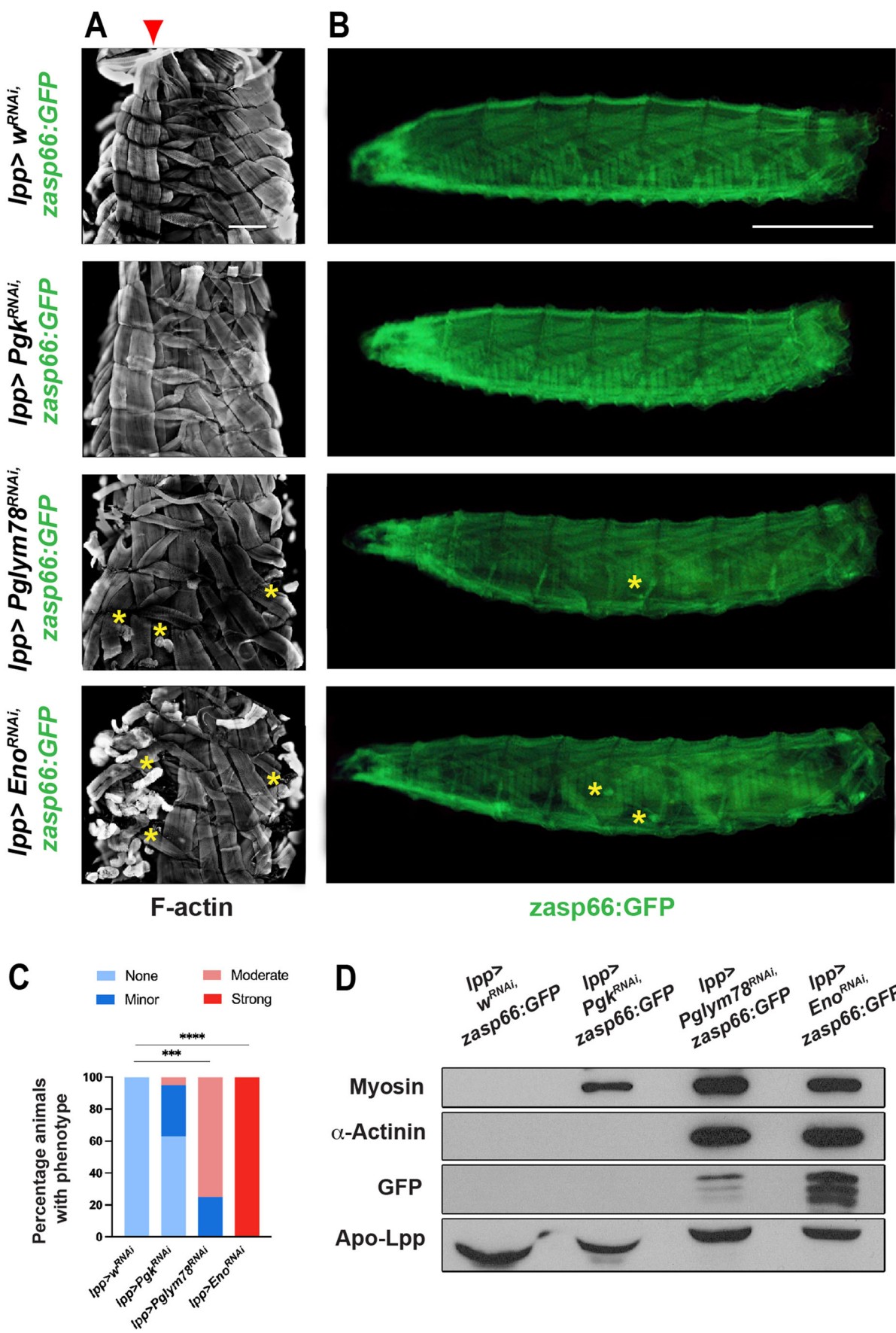

**Figure 3. Fat body *Pglym78* or *Eno* knock-down promotes body wall muscle disorganization.**

(A, B) Larval body wall muscles after glycolytic enzymes knock-down in fat body cells are monitored using F-actin staining (white in **A**) on dissected fixed larvae, or the Zasp66:GFP reporter in live larvae (green in **B**). White bar: 200 μm in (**A**) and 750 μm in (**B**). Red arrowhead in (**A**) indicates the VL3/VL4 muscles. Yellow stars highlight damaged muscles. In all panels anterior is up (**A**) or left (**B**). (**C**) Quantification of the muscle defects from (**A**) and categorized according to severity (see Fig. EV3A). For the different genotypes, n > 11. Chi-square test, ****p < 0.0001, ***p = 0.001. (**D**) Western blot of whole protein extracts of hemolymph (5 μl samples) from the larvae shown in (**A'**) and monitoring the presence of muscle proteins such as Myosin, α-Actinin, or Zasp66 (GFP). Apo-LppII is used as a loading control. Source data are available online for this figure.

also reduced muscle disorganization in *lpp-Gal4> Pglym78-RNAi* animals, with more animals exhibiting only minor defects than on normal food (Fig. EV3D–F). This suggests that rescued fat body cells prevent muscle disorganization. However, as noted earlier, we cannot exclude that the supplemented pyruvate acted independently of its effects on adipocytes, for instance, through other cell types or directly on muscles to prevent their disorganization.

Together, these results show that impairing glycolysis in the adipose cells in the fat body of *Drosophila* larvae triggers (1) adipocyte atrophy and (2) a non-autonomous distant muscle wasting, for which the release of α-Actinin in the blood could serve as a marker.

## The mTOR effector REPTOR is required for fat body atrophy and distant muscle wasting

The mTOR kinase and the TORC1 complex, represent one of the major integrator and key regulator of cellular metabolism. Typically, in rich conditions and in particular, under high amino-acid supply, the mTOR pathway is activated, promoting anabolic reactions (Bjedov and Rallis, 2020). Upon glycolysis enzyme knock-down, we observed low TORC1 activity. First, using qRT-PCR, we observed a major downregulation of the expressions at the RNA level of the mTor kinase and of the key TORC1 activator Rheb (Garami et al, 2003; Saucedo et al, 2003; Stocker et al, 2003; Zhang et al, 2003) in *lpp-Gal4> Pglym78-RNAi* fat body cells, suggesting that these cells were less prone to activate TORC1 (Fig. 4A). Second, the ribosomal activity regulator S6k was hypo-phosphorylated in the fat body of *lpp-Gal4> Pglym78-RNAi* animals compared to controls, further suggesting low TORC1 activity (Fig. 4B; (Saitoh et al, 2002)). Alternatively, this low phospho-S6k could be the consequence of higher phosphatase activity, but when combined with low *Rheb* expression, this result strongly suggests that TORC1 activity is impaired in the adipocytes of *lpp-Gal4> Pglym78-RNAi*.

During metabolic control in *Drosophila*, TORC1 acts mainly through its negative regulation of the transcription co-factor REPTOR. TORC1 phosphorylates REPTOR, thus promoting its cytoplasmic retention. When TORC1 activity is low, hypo-phosphorylated REPTOR translocates to the nucleus to turn on the transcription of target genes such as *4E-BP/thor* (translation inhibition), or *unk* (autophagy) (Tiebe et al, 2015). In *lpp-Gal4> Pglym78-RNAi*, low TORC1 signaling should thus translate into a higher REPTOR activity. However, even though mTOR activity was decreased in *Pglym78-RNAi* cells, we observed reduced REPTOR activity as evidenced by low expression levels of the REPTOR transcriptional targets *unk* and *4E-BP/thor* (Fig. 4C; (Tiebe et al, 2015)). In *Pglym78-RNAi* fat body cells, we observed a robust decrease in the expression of *REPTOR-BP*, the transcriptional

factor partner of REPTOR and required for REPTOR transcriptional output (Fig. 4C). Our results thus suggest that upon glycolysis blockade, even though mTOR activity is low, REPTOR transcriptional activity remains low, probably due to decreased REPTOR-BP expression. But, even though weakened, REPTOR was still active in *Pglym78-RNAi* cells since removing *REPTOR* by RNAi led to an even stronger decrease in the expression of *unk* and *4E-BP* (Fig. 4D).

Strikingly, even though REPTOR transcriptional activity was lower in *Pglym78-RNAi* cells compared to controls, knocking down *REPTOR* by RNAi in adipocytes very efficiently suppressed the fat body cells atrophy, and the average size of adipocytes was restored (compared to *Pglym78* RNAi alone; Fig. 4E,G,H). This result suggests that the remaining REPTOR activity in *Pglym78-RNAi* cells contributed to their atrophy. Alternatively, since RNAi driven by *lpp-G4* occurs from early larval life, it is possible that the effects observed represent an earlier requirement for *REPTOR* during adipose tissue development. Consistently, fat body *REPTOR* RNAi also suppressed distant muscle wasting, as evidenced by properly shaped muscles and the absence of severed muscles by whole muscle prep (Fig. 4F,J). Similarly, the intensity of F-actin staining in muscles, and in particular in the VL3 and VL4 muscles were restored, and α-Actinin protein levels in the circulating hemolymph were reduced (Fig. 4I).

Taken together, these results show that upon glycolysis shut down, adipocyte atrophy is mediated, at least in part, by the remaining REPTOR activity, leading to distant muscle disorganization.

## Upd3 is expressed by adipose tissue but is not required nor sufficient for muscle wasting

We then asked what could be the molecular link that would mediate this inter-organ effect of adipose tissue atrophy on muscles. Adopting a candidate approach, we first investigated the potential role of upd3. Indeed, in a *Drosophila* adult model of cachexia and muscle wasting, upd3 has been shown to be secreted by yki-driven intestinal tumors and then act on distant muscles to command wasting (Ding et al, 2021). Furthermore, in a larval model of neoplastic *RasV12/scrib-* eye disc tumors, impairing the upd3 receptor domeless in body wall muscles significantly weakened muscle wasting and disorganization (Hodgson et al, 2021). These different arguments suggest that elevated upd3 levels could be an important mediator of muscle wasting, at least in the context of cachectic tumors.

Monitoring RNA expression levels by qRT-PCR, revealed that *upd3* was upregulated in the atrophic fat body of *lpp-Gal4> Pglym78-RNAi* animals (Fig. EV4A). It should be noted however that absolute levels might be low (high number of qPCR cycles

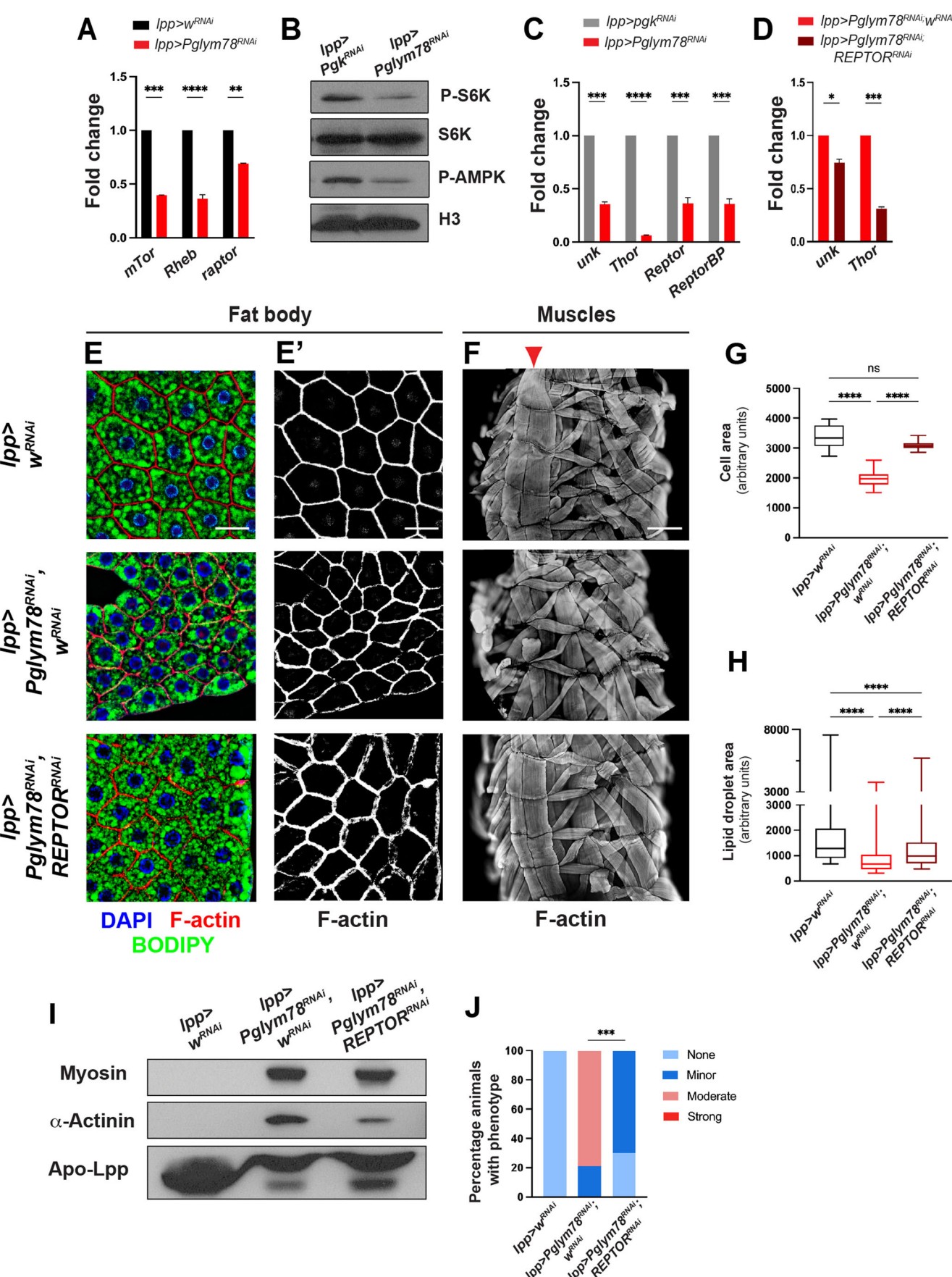

◄ **Figure 4. The mTOR target *REPTOR* mediates the fat atrophy and distant muscle disorganization in fat body *Pglym78* knock-down larvae.**

(A) Expression of the TORC1 components *mTor* and *raptor* and of the main regulator *Rheb* in fat body cells monitored by qRT-PCR. Reference gene for normalization: *rp49/RpL32*. Biological replicates n = 3. Error bars show the standard error of the mean (sem). One-way ANOVA statistical test. ****p < 0.0001, ***p = 0.0001, **p = 0.0026. (B) Western blot of whole protein extracts of fat body tissues monitoring the TORC1 phosphorylation target S6k (P-S6k) and the activity of AMPK (P-AMPK). Total S6k and Histone H3 are used as loading control. (C) Expression of the REPTOR target genes *4E-BP/thor* and *unk* and of the REPTOR transcriptional complex components *REPTOR* and *REPTOR-BP* in fat body cells monitored by qRT-PCR. Reference gene for normalization: *rp49/RpL32*. Biological replicates n = 3. Error bars show the standard error of the mean (sem). Biological triplicates. one-way ANOVA statistical test. ****p < 0.0001, ***p = 0.0001. (D) Expression of *4E-BP/thor* and *unk* in *Pglym78-RNAi* fat body cells monitored by qRT-PCR after *REPTOR* invalidation. Reference gene for normalization: *rp49/RpL32*. Biological replicates n = 3. Error bars show the standard error of the mean (sem). One-way ANOVA statistical test. ***p = 0.001, *p = 0.0240. (E) Fat body staining of the indicated genotypes showing nuclei (DAPI, blue in (E)), cell cortex (F-actin, red in (E) and white in (E')), and lipid droplets (BODIPY, green in (E)); white bar 50 μm. (F) Larval body wall muscles of the same genotypes as in (E) monitored using F-actin staining (white in (F)) on dissected fixed larvae; white bar 200 μm. The red arrowhead indicates the VL3/VL4 muscles. In all panels anterior is up. (G, H) Quantification of cell size as a measure of fat body cell atrophy (G) and lipid droplets content (H) from (E), and represented as box plots where the whiskers represent the maxima and minima experimental points, the boxes represent the 25 and 75% percentiles, and where the center line is the median. Biological replicates n = 10. One-way ANOVA (Kruskal–Wallis) statistical test, ****p < 0.0001, ns not significant. (I) Western blot of whole protein extracts of hemolymph (5 μl samples) from the larvae shown in (E, F) and monitoring the presence of muscle proteins such as Myosin and α-Actinin. Apo-LppII is used as a loading control. (J) Quantification of the muscle defects from (F) and categorized according to severity (see Fig. EV3A). For the different genotypes, n > 10. Chi-square test, ***p = 0.001. Source data are available online for this figure.

required for detection), and that the fold increase in *upd3* expression observed in the fat body of *lpp-Gal4> Pglym78-RNAi* animals stemmed from the absence of *upd3* expression in control animals. To formally test the potential function of fat body-derived upd3, we first performed loss-of-function experiments. We did not observe any rescue of the fat body atrophy nor of the muscle disorganization and α-Actinin release in the blood, when driving in *lpp-Gal4> Pglym78-RNAi* larvae an RNAi directed against *upd3* (Fig. EV4B–F). Even though the depletion of *upd3* by RNAi was only partial (here circa 70%, Fig. EV1A), these results suggest that upd3, produced by adipose cells after glycolysis shut-down, was not required for distant wasting of muscles. However, it remained possible that the critical source of upd3 were not the adipocytes (targeted by *lpp-Gal4*), but were the fat body resident hemocytes, and that upd3 would be required in these cells for muscle wasting. To address this possibility, we used the *Cg-Gal4* driver which drives expression in both adipocytes and hemocytes. Driving *Pglym78-RNAi* alone using *Cg-Gal4* led to similar fat body atrophy and distant muscle wasting as observed with *lpp-Gal4* but, here again, we did not observe any rescue of the muscle wasting phenotype by targeting *upd3* depletion in both adipocytes and hemocytes (Fig. EV5A–C). While, we cannot rule out that *Pglym78-RNAi*-mediated glycolysis shut-down in both adipocytes and hemocytes could trigger a different response and inter-organ messengers, the lack of modification of distant muscle wasting using *lpp-Gal4* and *Cg-Gal4* suggests that it was not mediated by *upd3* expression in the fat body (adipocytes and hemocytes).

We next performed gain-of-function experiments to test whether upd3 could be sufficient to induce muscle wasting. We thus overexpressed upd3 using either a *UAS-upd3* construct previously used (Romão et al, 2021) or a fly line in which *upd3* overexpression is achieved through the expression of a UAS-controlled dCas9-SAM, combined with appropriate sgRNA located in the upstream vicinity of the *upd3* gene (flySAM2.0 upd3; (Jia et al, 2018)). Although high expression of *upd3* could be achieved with the different tools used, the overexpression of upd3 in adipocytes (*lpp-Gal4*), or in the adipocytes and hemocytes (*Cg-Gal4*) did not promote muscle wasting, nor α-Actinin release in the blood (Fig. EV5D–G). Taken together, these results strongly suggest that *upd3* upregulated in a fat body undergoing atrophy after glycolysis shut down did not mediate distant muscle disorganization.

## TNF-α/egr signaling is required for adipose-mediated muscle wasting

Having ruled out upd3, we then tested whether TNF-α/egr could be mediating the effects of the fat body atrophy on distant muscle wasting. Indeed, even though the link between TNF-α signaling and muscle wasting is complex, TNF-α has long been associated with inflammation, cachexia, and sarcopenia in patients (Patel and Patel, 2017). Recent studies on *Drosophila* larval models of tumor-induced cachexia showed that knocking down in muscles wengen, one of the fly TNF-α receptor, partly suppressed muscle disorganization and wasting (Hodgson et al, 2021).

We first monitored the expression of *egr* and *Tace*, the gene coding for a metalloprotease required for the processing and release of the soluble active form of egr. The expression of *egr* was downregulated, but the expression of *Tace* was upregulated at the RNA level following glycolysis shutdown (Fig. 5A), suggesting that even though the *egr* gene was less expressed, there might be increased levels of active egr. Indeed, we detected an increase in soluble active egr protein in the hemolymph of *lpp-Gal4> Pglym78-RNAi* animals compared to controls at 5d ael. Importantly, this was abolished in *REPTOR-RNAi* animals (Fig. 5B). However, since the expressions of *egr* and *Tace* were not affected by *REPTOR-RNAi*, confirming earlier reports (Fig. 5A; (Agrawal et al, 2016)), the control of egr circulating levels by REPTOR might result from changes to Tace metalloprotease activity and/or egr secretion.

We then tested whether egr signaling was required for the distant muscle wasting by interfering with the expressions of *egr* and *Tace*. Expressing RNAi against *egr* or *Tace* significantly rescued the muscle disorganization observed in *lpp-Gal4> Pglym78-RNAi* animals, as evidenced by normal F-actin staining (in particular at the level of the VL3 and VL4 muscles) and morphology of muscle fibers, without any bright accumulation, and a significant decrease in α-Actinin release in the blood (Fig. 5D,F–H). Importantly, this rescue of the muscles was observed without significant change to the fat body, which remained atrophied (Fig. 5C,E). But if egr produced by adipocytes was required for distant muscle wasting, it was not sufficient. Indeed, the overexpression of an active egr did not promote muscle wasting or α-Actinin release in the blood (Fig. 5I), suggesting that the action of egr is dependent on context, and that egr represents a permissive rather than an instructive

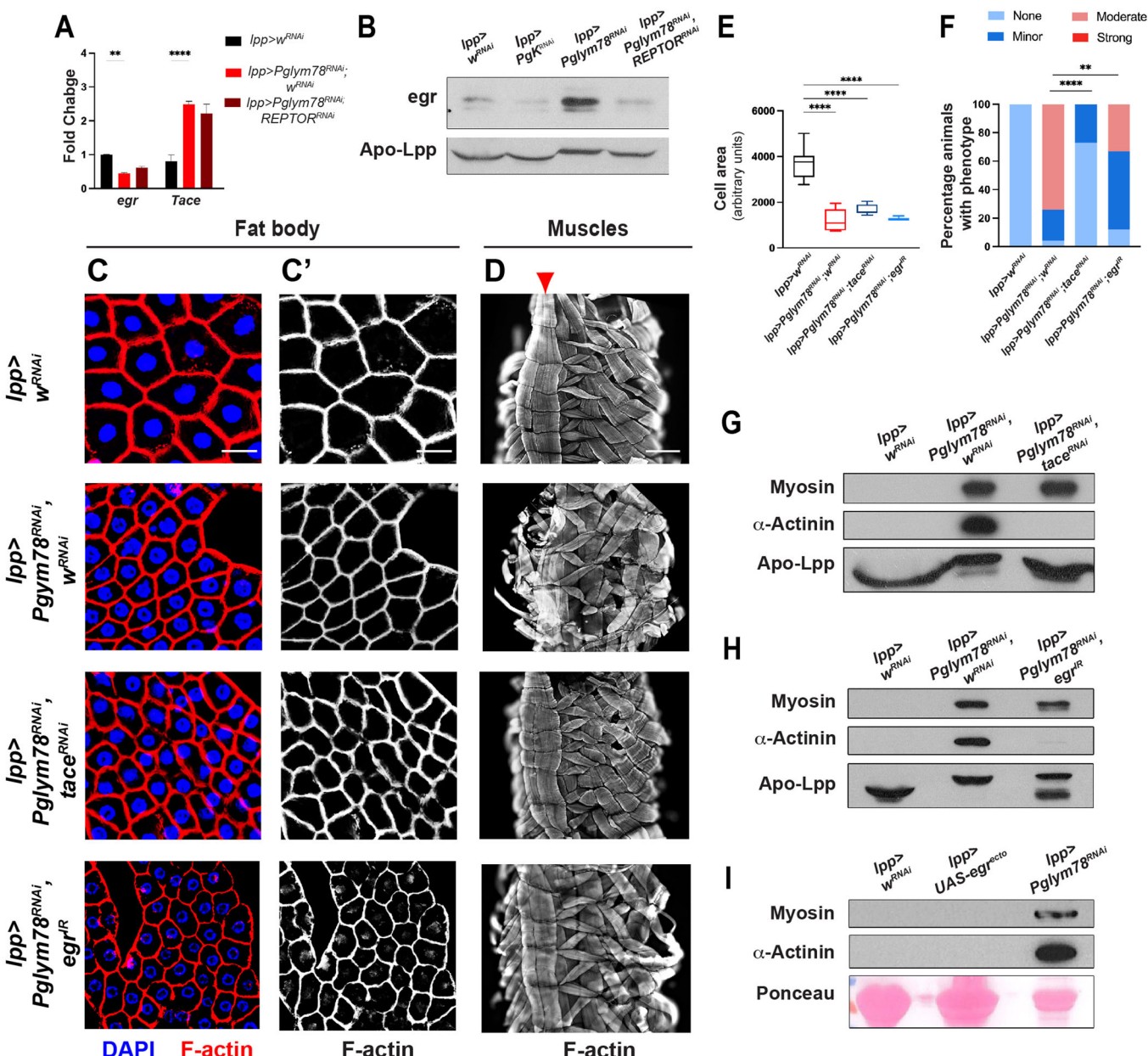

**Figure 5. Egr produced by the atrophied fat body is required for distant body wall muscle disorganization.**

(A) Expression of the TNF-α pathway members *egr* and *Tace* in fat body cells after *Pglym78* knock-down or *Pglym78* and *REPTOR* double knock-down monitored by qRT-PCR. Reference gene for normalization: *rp49/RpL32*. Biological replicates *n* = 3. Error bars show the standard error of the mean (sem). One-way ANOVA statistical test. ****p < 0.0001, **p = 0.0010. (B) Western blot of whole protein extracts of hemolymph (5 μl samples) from the larvae of the indicated genotypes and monitoring the presence of secreted egr protein. Apo-LppII is used as a loading control. (C) Fat body staining of the indicated genotypes showing nuclei (DAPI, blue in (C)), cell cortex (F-actin, red in (C), and white in C'); white bar 50 μm. (D) Larval body wall muscles of the same genotypes as in (C) monitored using F-actin staining (white in D) on dissected fixed larvae; white bar 200 μm. The red arrowhead indicates the VL3/VL4 muscles. In all panels anterior is up. (E) Quantification of cell size as measure of fat body cell atrophy from (C), and represented has box plots where the whiskers represent the maxima and minima experimental points, the boxes represent the 25 and 75% percentiles, and where center line is the median. Biological replicates *n* = 6–9. One-way ANOVA statistical test, ****p < 0.0001. (F) Quantification of the muscle defects from (D) and categorized according to severity (see Fig. EV3A). For the different genotypes, *n* > 11. Chi-square test ****p < 0.0001, **p = 0.0028. (G, H) Western blot of whole protein extracts of hemolymph (5 μl samples) from the larvae shown in (C, D) and monitoring the presence of muscle proteins such as Myosin and α-Actinin. Apo-LppII is used as a loading control. (I) Western blot of whole protein extracts of hemolymph (5 μl samples) from the indicated genotypes and monitoring the presence of muscle proteins such as Myosin and α-Actinin. Ponceau S staining showing the Lsp proteins is used as a loading control. Source data are available online for this figure.

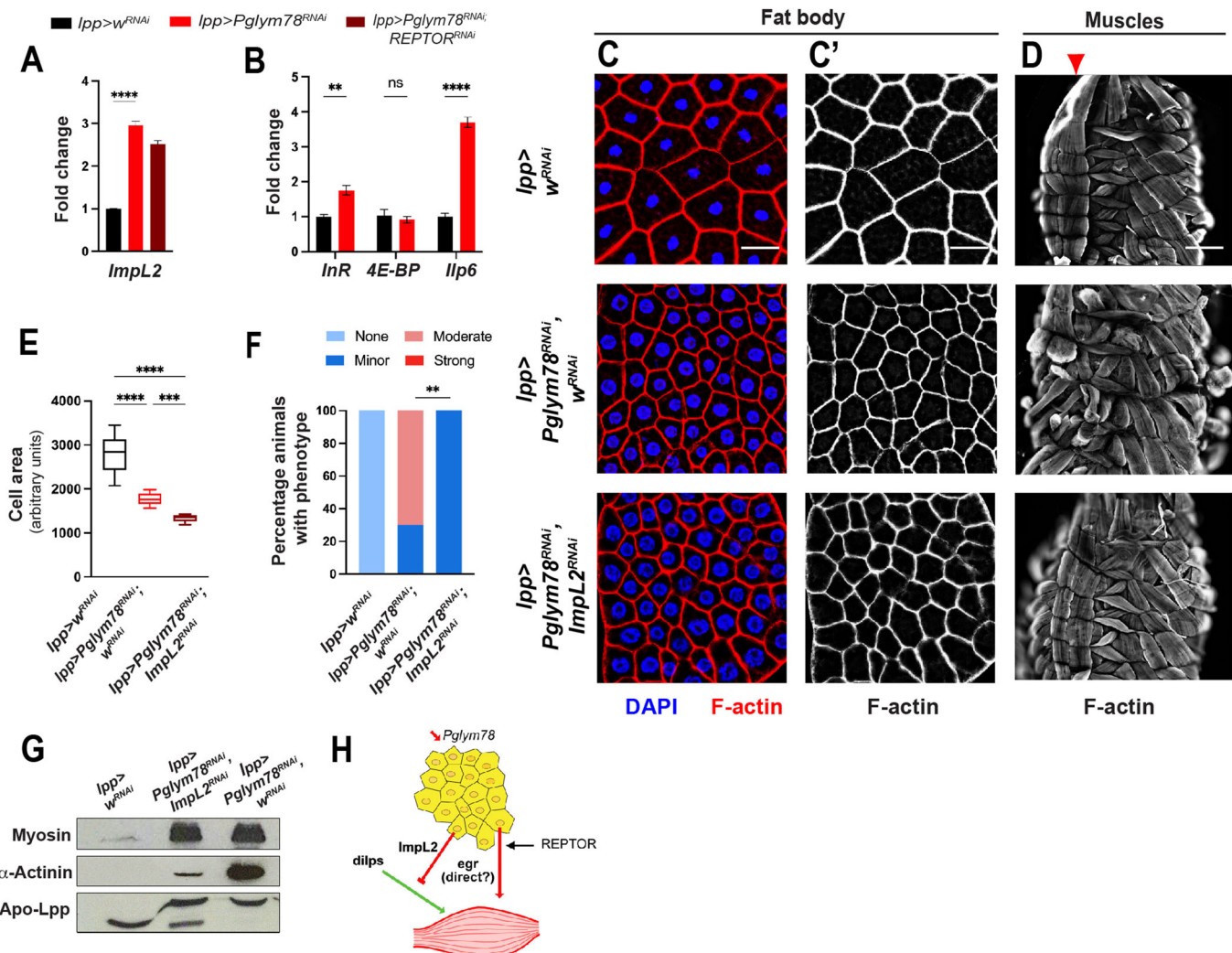

**Figure 6. The extracellular insulin trap ImpL2 produced by the atrophied fat body is required for distant body wall muscle disorganization.**

(A) Expression of the extracellular insulin trap *ImpL2* in fat body cells after *Pglym78* knock-down or *Pglym78* and *REPTOR* double knock-down monitored by qRT-PCR. Reference gene for normalization: *rp49/RpL32*. Biological replicates $n = 3$. Error bars show the standard error of the mean (sem). One-way ANOVA statistical test. ****$p < 0.0001$. (B) Expression of Insulin pathway transcriptional targets *InR*, *4E-BP/thor*, and *dilp6/Ilp6* in carcasses mainly composed of body wall muscles from animals with *Pglym78* knock-down in fat body cells and monitored by qRT-PCR. Reference gene for normalization: *rp49/RpL32*. Biological replicates $n = 3$. Error bars show the standard error of the mean (sem). One-way ANOVA statistical test. ****$p < 0.0001$, **$p = 0.0010$, ns not significant. (C) Fat body staining of the indicated genotypes showing nuclei (DAPI, blue in C), cell cortex (F-actin, red in C), and white in C'); white bar 50 μm. (D) Larval body wall muscles of the same genotypes as in (C) monitored using F-actin staining (white in D) on dissected fixed larvae; white bar 200 μm. Red arrowheads indicate the VL3/VL4 muscles. In all panels anterior is up. (E) Quantification of cell size as measure of fat body cell atrophy from (C), and represented has box plots where the whiskers represent the maxima and minima experimental points, the boxes represent the 25 and 75% percentiles, and where center line is the median. Biological replicates $n = 10–14$. One-way ANOVA (Kruskal–Wallis) statistical test, ****$p < 0.0001$, ***$p = 0.0010$. (F) Quantification of the muscle defects from (D) and categorized according to severity (see Fig. EV3A). For the different genotypes, $n = 10$. Chi-square test, **$p = 0.0034$. (G) Western blot of whole protein extracts of hemolymph (5 μl samples) from the larvae shown in (C, D) and monitoring the presence of muscle proteins such as Myosin and α-Actinin. Apo-LppII is used as a loading control. (H) Model for the fat body to muscle communication after adipose tissue glycolysis knock-down. Source data are available online for this figure.

signal, or that egr must cooperate with other signals to promote distant muscle disorganization.

## Muscle wasting is triggered by low Insulin signaling

One obvious candidate for the context-dependent action of egr, is Insulin signaling. Indeed, previous studies showed that larval muscle size and organization is supported by insulin receptor and Foxo activities (Demontis and Perrimon, 2009). In *Drosophila* larvae, the insulin-producing cells in the larval brain represent the main source of circulating insulin-like peptides (dilps). However, the activity of these dilps is further controlled by secreted molecules that can bind and trap dilps, in particular, ImpL2 (Alic et al, 2011; Bader et al, 2013; Honegger et al, 2008). *ImpL2* expression was actually highly upregulated in fat body cells from *lpp-Gal4> Pglym78-RNAi*, and appeared independent of *REPTOR* (Fig. 6A).

We reasoned that this increase in ImpL2 could dampen circulating dilps, and thus block Insulin signaling in peripheral tissues, and more particularly in muscles. Low circulating insulin levels were supported by the observation that *lpp-Gal4> Pglym78-RNAi* animals showed transcriptional characteristics of low insulin signaling with increased expression levels of *InR* and *Ilp6* in body-wall muscles (Fig. 6B; (Sriskanthadevan-Pirahas et al, 2022)).

To test the relevance of *ImpL2* expression in the fat body on the distant muscle disorganization, we invalidated it by RNAi. Inhibiting *ImpL2* in adipocytes (*lpp-Gal4*) led to a dramatic rescue of the distant muscles: we observed better formed body wall muscles (with increased F-actin staining in VL3 and VL4 muscles in particular), and lower α-Actinin release in the blood (Fig. 6D,F,G). Similarly to what was observed for egr/Tace signaling, the muscle rescue by *ImpL2* knock-down occurred without any suppression of the fat body atrophy (Fig. 6C,E).

We were not able to test whether ImpL2 overexpression was sufficient for muscle disorganization since its overexpression using available UAS constructs led to a dramatic developmental delay and tiny larvae (Bader et al, 2013). Similarly, the double overexpression of ImpL2 and egr in fat body cells resulted in early developmental defects, preventing us from assessing whether their combined over-expressions could phenocopy the invalidation of *Pglym78* on distant muscle disorganization.

Taken together, these results support a model in which glycolysis shut-down in adipocytes leads to their atrophy and secretion of egr, at least in part through processes controlled by REPTOR. In parallel, glycolysis impairment controls the production of ImpL2 and potentially additional factors, which cooperate with egr to trigger distant muscle disorganization (Fig. 6H).

## Discussion

In this study, we showed that impairing glycolysis in adipocytes through the downregulation of Pglym78 or Eno enzymes, led to dramatic atrophy of the lipid reserves, reminiscent of previous reports linking glucose metabolism and lipid storage in mammals and *Drosophila* (Ding et al, 2018; Garrido et al, 2015; Heier and Kühnlein, 2018). First, glucose metabolism and glycolysis have been linked to de novo lipogenesis and fatty acid and lipid production. Glycolysis produces pyruvate, which is converted to Ac-CoA, fueling the TCA cycle in mitochondria. Citrate from the TCA cycle can exit the mitochondria and serves to generate fatty acids through the de novo lipogenesis pathway and the action of the FASN and ACC enzymes. Even though, fat body cells can store dietary lipids, they also rely on de novo lipogenesis for lipid droplets formation (Garrido et al, 2015; Heier and Kühnlein, 2018; Parvy et al, 2012; Song et al, 2018). The activity of FASN requires NADPH, which can be produced, amongst other sources, by the pentose pathway, which branches off from glycolysis. Second, glycolysis is also connected to lipid consumption. Indeed, when the glycolysis flux is limited, cells can use fatty acid β-oxidation, instead of pyruvate, to produce Ac-CoA, leading to lipid consumption (McMullen et al, 2023). Measuring steady-state concentrations of several metabolites, suggested that impairing glycolysis at the level of Pglym78, pushed metabolic reprogramming and adaptation. Indeed, we measured lower triglycerides, as well as increased ATP and pyruvate concentrations, which are consistent with a shift from

glycolysis and pyruvate consumption (accumulation of pyruvate) towards β-oxidation and lipid consumption (less triglycerides). The relative contributions of increased consumption (β-oxidation), and decreased production (de novo lipogenesis), in the lipid reserve shrinkage remain to be explored.

Looking at the potential global effects of the glycolysis blockade in fat body cells, we uncovered a link between fat body atrophy and body wall muscle disorganization in *Drosophila* larvae. We further showed that this inter-organ communication relies on the cooperation of at least two signals: (i) TNF-α/egr release by atrophic adipocytes and (ii) inhibition of circulating insulin signaling by ImpL2 (Fig. 6H). These two signals produced by the fat body were required to control muscle disorganization, but other yet unidentified signals might participate in this adipose> muscle communication. In this glycolysis shut-down fat body atrophy model, we could not find any involvement of IL-6/upd3 in the disorganization of muscles, unlike what had been shown in the case of tumor-induced muscle wasting in adult *Drosophila* cachexia models (Ding et al, 2021). It should be noted that, while we ruled out fat body-derived upd3, we did not formally test whether circulating upd ligands secreted by other organs (in response to the atrophied fat body) or whether Jak/Stat signaling in muscles played any role in muscle disorganization induced by fat atrophy.

TNF-α/egr signaling was activated upon glycolysis shut-down in larval adipocytes, as evidenced by the elevated egr circulating in the hemolymph of *Pglym78-RNAi* animals. While the increased *Tace* expression suggests that more processed egr might be produced, the mechanisms behind this increased circulating egr protein levels remain to be explored, but they are controlled, at least in part, by REPTOR. As published previously, REPTOR did not control *egr* or *Tace* expressions, suggesting that REPTOR controls alternative mechanisms leading to egr secretion, which could involve higher *egr* mRNA translation leading to increased protein levels, stabilization of activated egr, or enhanced Tace cleaving activity. Reports have linked lipids such as ceramides and cholesterol in the control of TACE/ADAM17 activity, and TNF-α secretion (Hooper, 2005; Lamour et al, 2011), and it would be interesting to explore whether REPTOR activity might influence the types and levels of lipids presents in membranes which could lead to differences in Tace activity. Whether egr can act directly on muscles, as suggested by the role of its receptor wengen during tumor-induced muscle wasting (Hodgson et al, 2021), or indirectly, or through a combination of direct and indirect effects, remains, however to be determined.

Since the overexpression of egr in fat body cells could not recapitulate the muscle disorganization observed upon *Pglym78-RNAi*, it suggests that egr signaling while required, must cooperate with other signals. One such signal is represented by the extracellular dilps inhibitor ImpL2 (Alic et al, 2011; Honegger et al, 2008). *ImpL2* was indeed required for the distant muscle wasting when knocked down in adipocytes, suggesting that the extracellular trapping of dilps and thus dampened Insulin signaling was a major contributor to muscle disorganization in this system. This could reflect muscle growth problems since Insulin receptor and Foxo signaling are required for the growth of larval body wall muscles, including the VL3 and VL4 muscles (Demontis and Perrimon, 2009). The critical role of extracellular Insulin inhibitors in muscle wasting was demonstrated previously in adult *Drosophila* cachexia models, in which either rasV12-driven tumorous larval

disks transplanted in adult flies, or yorkie-driven adult midgut tumors, induced muscle disorganization and wasting through the secretion of ImpL2 (Figueroa-Clarevega and Bilder, 2015; Kwon et al, 2015). Our results indicate that, like adult muscles, larval muscles are sensitive to ImpL2 levels, at least in certain conditions.

We could show that in response to *Pglym78* knock-down, mTOR activity was inhibited, consistent with the nutritional challenge facing adipocytes. However, the mechanisms by which, glycolysis impairment after *Pglym78* knock-down controls TORC1 activity remain fragmentary. Our results showed that glycolysis shut-down resulted in low *Rheb* expression. Since Rheb is required for TORC1 activation, this low *Rheb* expression might contribute to the low TORC1 activity, but other mechanisms might also act in parallel. Alternatively, AMPK signaling has been shown to antagonize mTOR signaling by phosphorylating and activating TSC2 (Inoki et al, 2006), or by phosphorylating and inhibiting raptor (Gwinn et al, 2008). However, AMPK is unlikely to be involved here since we observed increased ATP levels and decreased levels of the activated p-AMPK (Fig. 4B). Earlier reports showed that mTOR pathway inhibition in fat body cells, resulted in systemic effects with smaller overall organisms, in part through strong retention of dilps in the IPCs and thus lower peripheral insulin signaling (Colombani et al, 2003; Delanoue et al, 2016; Géminard et al, 2009). The tools used in these earlier studies (knock-down of the amino-acid transporter slimfast, or of the mTOR activator Rheb, or overexpression of the mTOR inhibitor TSC1/2) led to a strong mTOR inhibition. After *Pglym78* RNAi, we only observed subtle effects on the overall growth of the larva with slightly smaller wing disks but larvae with normal size (see Fig. EV2). These results suggest that at the small level of mTOR inhibition achieved here by *Pglym78* RNAi, the remote control of organismal growth by mTOR sensing in the adipose tissue, is likely negligible. However, we cannot exclude that it might contribute to the distant muscle disorganization observed.

But even though TORC1 activity was inhibited, that did not result in higher REPTOR activity, the main transcriptional effector downstream of mTOR (Tiebe et al, 2015). Indeed, we could observe lower expression of REPTOR target genes such as *4E-BP/thor* and *unk* (Guertin et al, 2006; Tiebe et al, 2015). This was likely mediated by lower expression of the REPTOR partner REPTOR-BP. Since REPTOR and REPTOR-BP have been implicated in implementing the mTOR transcriptional programs in response to nutrient stress and starvation, the low REPTOR/REPTOR-BP activity in atrophied *Pglym78-RNAi* cells could be interpreted as a homeostatic response and consequence of the atrophy in order to prevent deleterious final exhaustion of the remaining reserves in adipocytes, or to ensure sufficient anabolic activity.

However, even if REPTOR transcriptional activity was down in *Pglym78-RNAi* cells, it was completely absent, and *REPTOR* inhibition in the fat body, suppressed fat body atrophy, suggesting thus that the remaining REPTOR activity is required for fat body atrophy, and might control metabolic reprogramming similarly to what was recently shown in cachectic adult *Drosophila* muscles (Saavedra et al, 2023). *REPTOR-RNAi* in fat body cells also suppressed distant muscle disorganization, suggesting that REPTOR is required for the production of signals affecting muscles, including egr. Glycolysis impairment controls the production of additional factors, which cooperate with egr to affect muscles, such as ImpL2. These additional signals, dependent or not on REPTOR, remain to be identified, and

could include other adipokines or specific trophic factors produced by the fat body. The atrophied fat body might indeed be unable to provide the nutrients necessary for muscle growth and/or maintenance of healthy muscles. This trophic stress, combined with the action of egr and ImpL2 would then lead to muscle disorganization (Fig. 6H). This model is supported by a recent study proposing that in a *Drosophila* larval model of tumor-induced cachexia, the atrophic fat body fails to export critical extracellular matrix components such as collagen IV/ viking, thus contributing to muscle wasting (Bakopoulos et al, 2023), or by the description in mammals of adipose tissue-derived signaling lipids promoting efficient insulin signaling in skeletal muscles (Cao et al, 2008). Further studies are needed to document the exchanges between the atrophied fat body and the affected muscles.

A recent report analyzing the role of adipocyte mitochondrial function in the regulation of organismal metabolism, showed that knocking down *TFAM*, the master transcription factor controlling mitochondrial genes, led to animals with increased general metabolism, which developed faster than controls, shortening larval life. Interestingly, in these animals, the fat body cells produced less egr and less ImpL2 leading to an overall putative higher insulin signaling and faster metabolism (Sriskanthadevan-Pirahas et al, 2022). This situation, impairing mitochondrial function, appears thus the opposite of the one we describe here, knocking down glycolysis. Whether TFAM and mitochondrial function are involved in the effects of glycolysis shut down or vice versa, remains to be explored. Further studies should help clarify the link between glycolysis, mitochondrial function, REPTOR, and TORC1 regulation in adipocytes, and how this results in adipokine secretion and the control of developmental timing, and muscle maintenance.

# Methods

## *Drosophila* genetics

All crosses were cultured at 25 °C on standard food, except for the experiments with food supplemented with 1% of sodium pyruvate (Sigma Aldricht #P5280).

The stocks used are listed below:
Bloomington *Drosophila* Stock Center: BDSC
Vienna *Drosophila* Resource Center: VDRC

| Name / Gene | Transgene | Integration | Stock / Origin |
|---|---|---|---|
| **Gal4 drivers** | | | |
| Cg-Gal4 | | on II | BDSC #7011 |
| Lpp-Gal4 | | on X | Gift from P. Léopold |
| Ppl-Gal4 | | on X | Gift from P. Léopold |
| **UAS RNAi lines** | | | |
| Eno | HMC05514 | attP40 | BDSC #64496 |
| egr | eiger.IR | on III | BDSC #58993 |
| ImpL2 | HMC03863 | attP40 | BDSC #55855 |
| Pgk | HMS00030 | attP2 | BDSC #33632 |
| Pglym78 | HMC04818 | attP40 | BDSC #57503 |
| REPTOR | JF02005 | attP2 | BDSC #25983 |

| Name / Gene | Transgene | Integration | Stock / Origin |
|---|---|---|---|
| Tace | KK109440 | VIE-260B | VDRC v106335 |
| upd3 | HMS00545 | attP2 | BDSC #33680 |
| w | HMS00045 | attP2 | BDSC #33644 |
| y | HMC05546 | attP40 | BDSC #64527 |
| y | KK104196 | VIE-260B | VDRC v106068 |
| UAS overexpression | | | |
| SAMdCas9-egr | GS03184 | attP40 | BDSC #81331 |
| SAMdCas9-upd3 | GS03275 | attP40 | BDSC #81371 |
| UAS-mCD8-GFP | | on III | BDSC #5130 |
| UAS-egr ecto | | on III | Gift from P. Léopold |
| UAS-ImpL2 | | on II | Gift from H. Stocker |
| UAS-upd3 | | | Gift from M. Milán |
| Reporter | | | |
| Zasp66-GFP | ZCL0663 | on III | BDSC #6824 |

## Pupariation curve

Females were put to lay eggs for 4 h, before removing all adults. The number of larvae that entered pupariation were counted at different time points after egg laying: 108, 114, 120, 131, 138, 145, 152, 161, 169, and 183 h. The percentage of pupariation was calculated considering the total number of larvae that finally reached the pupal stage.

## Hemolymph extraction and trehalose measurement

Third instar larvae were washed in water and then in 70% EtOH before being dried on filter paper. Groups of ten larvae were placed on a piece of Parafilm on ice. Hemolymph was extracted by piercing carefully the cuticle with forceps and then collected by pipetting. For Western blot analyses, 5 µl of collected hemolymph were immediately complemented by 5 µl of 2X Laemmli buffer. 1.5 µl of hemolymph was used to measure the glucose and trehalose according to the Glucose assay kit (Sigma-Aldrich #GAGO20). Since similar volumes were collected between the different genotypes, these measures provide the relative concentrations without needing further normalization. To remove cells and tissue debris, the hemolymph was centrifuged at 4 °C for 30 min at 1000×g, and the supernatant was further centrifuged at 4 °C for 20 min at 15000×g.

## Metabolites measurements

For metabolite measurements, the approaches differed whether we measured on whole animals or on dissected tissues. Since the fat body is the main organ storing triglycerides (TAG) and glycogen in the developing larvae, approaches on whole larvae could bring an approximation of the quantity stored in the fat body. Here the normalization was done by the number of animals. Measurements were done on pools of five larvae, washed in cold PBS, whipped, and flash-frozen in liquid nitrogen, with four to six biological replicates. But we also performed measurements on dissected fat bodies for Pyruvate, ATP, and TAG. The *Pglym78-RNAi* fat body tissues were severely atrophied with many molecules affected,

including total protein content. We opted to normalize by the total nucleic acid content, which was the least likely to be affected in the atrophied adipocytes. There, measurements were performed on pools of fat bodies from five larvae dissected in cold PBS and flash-frozen in liquid nitrogen, with three to five biological replicates.

Frozen larvae or tissues where first mechanically homogenized using a mixer mill Retsch MM400 (Verder Scientific), and the concentrations of the different metabolites were measured using dedicated colorimetric or fluorescent detection kits, and followed the manufacturers' instructions. Glycogen levels were measured using the Glucose assay kit (Sigma-Aldrich #GAGO20) after converting glycogen into glucose. ATP, pyruvate, and TAG levels were measured using the ATP Bioluminescence Assay Kit CLS II (Roche #11699695001), the Pyruvate Assay kit (Sigma-Aldrich #MAK322), and the Serum Triglyceride Quantification kit (Cell Biolabs #STA-396), respectively.

## Immunofluorescence

Tissues were dissected in PBS and fixed at room temperature (RT) in 4% formaldehyde for 30 min. After washing 3x 20 min in PBT (PBS 0.2% Triton X-100), non-specific epitopes were blocked for 30 min in PBT-BSA (PBT 0.5% BSA). Primary antibodies were incubated overnight at 4 °C in PBT-BSA. Samples were then washed 3x for 20 min in PBT. Secondary antibodies were incubated for 2 h at RT in PBT-BSA. Samples were washed 3x in PBT and mounted in CitiFluor™ AF1 (Agar). Images were acquired on an upright Leica THUNDER microscope.

Primary antibodies used in this study: Mouse anti-α-actinin (1:25; Developmental Study Hybridoma Bank – DSHB #2G3-3D7), Rat anti-ECadherin (1:25; DSHB #DCAD2), Rabbit anti-GFP (1:200; Torrey Pines Biolabs #TP401). Secondary antibodies used conjugated to Alexa Fluor 488, 555, 647, or to Cy3 were from Jackson Labs Immuno Research (1:200). DAPI was used at 1:1000 and Phalloidin-Rhodamine (Sigma-Aldrich #P1951) was used at 1:200.

## Lipid droplets staining with BODIPY

Fat bodies from third instar larvae were dissected in PBS and fixed in 4% formaldehyde for 30 min at RT. Tissues were then rinsed 2x in PBS and incubated in BODIPY FL Dye (1:500 dilution from a 1 mg/mL stock, Thermo Fisher) for 30 min in PBS. They were rinsed 2x in PBS and immediately mounted in CitiFluor™ AF1 (Agar). Images were acquired on a Leica THUNDER microscope. Lipid droplet numbers and sizes were measured using Fiji (ImageJ) software (Schindelin et al, 2012).

## Cell area quantification

Cell segmentation was performed using Cellpose with the cyto2 model (Stringer et al, 2021), followed by quantification of cell area in Fiji (Schindelin et al, 2012) using the image segmentation masks generated.

## Muscle wasting quantification

About 10–15 larvae per genotype were classified according to muscle defects into four different categories (in ascending order of severity): None, Minor, Moderate, and Strong (Fig. EV3A). These categories are based on the weakness of the ventral and dorsal muscles from the

A1–A5 segments. The rupture of the myofibers is represented as dense and bright dots due to the accumulation of F-actin.

## Western blot

Western blot analyses were performed according to standard protocols. Primary antibodies used in this study: Mouse anti-Actin (1:50, DSHB #JLA20), Mouse anti-α-actinin (1:2000; DSHB #2G3-3D7), Mouse anti-egr (1:100; gift from K. Basler; unpublished), Mouse anti-myosin (1:2000; DSHB #3E8-3D3), Rabbit anti-p-AMPKα 40H9 (1:1000; CST #2535), Rabbit anti-lipophorin II (1:15000; gift from J. Culi), Rabbit anti-GFP (1:2000, Molecular Probes: A6455), anti-Histone H3 (1:100; CST #4499), Rabbit anti-phospho-Drosophila p70 S6 Kinase (1:100; CST #9209), Guinea-pig anti-Drosophila S6K (1:3000; gift from A. Teleman; (Hahn et al, 2010)). Secondary antibodies conjugated to HRP were from Jackson Labs Immuno Research (1:15000).

## Quantitative RT-PCR

For each tissue, pooled groups of ten dissected tissues were lysed chemically with QIAzol Lysis Reagent (QIAGEN) and physically using a mixer mill Retsch MM400 (Verder Scientific). Total RNA was then purified using QIAGEN kits (QIAGEN RNeasy Lipid #74804 or RNeasy Plus #74134). Genomic DNA was removed by incubating with DNase (QIAGEN #79254), and cDNA was retro-transcribed using SuperScript III (Invitrogen #18080-044). Semi-quantitative qPCR was performed on biological triplicates using SYBR Green I Master mix on a LightCycler® 480 (Roche). Fold change was estimated using the ΔΔCT approach.

Primers used:

| Gene name | Forward primer | Reverse primer |
| --- | --- | --- |
| Act5C | GAGCGCGGTTACTCTTTCAC | ACTTCTCCAACGAGGAGCTG |
| Eno | AGGCCATGAAGATGGGCTC | GCCTCCTTGTTGGACTGGATG |
| egr | GCATCCTCAGCCTCAAATGA | CCTGAAGCTCGTGTGTGATTTCC |
| Ilp6 | CCCTTGGCGATGTATTTCCCA | CTTGCAGCACAAATCGGTTAC |
| ImpL2 | AAGAGCCGTGGACCTGGTA | TTGGTGAACTTGAGCCAGTCG |
| InR | AAGCGTGGGAAAATTAAGATGGA | GGCTGTCAACTGCTTCTACTG |
| mTor | TTACTGCCAGGAAGGGCATTT | TGACGGACACATCGTTGATTAG |
| Pgk | ATCACCAGCAACCAGAGAATTG | TGCCAGGGTGTACTTGATGTT |
| Pglym78 | AGTCCGAGTGGAACCAGAAGA | CCTCCTCCTGACCCTTTTCG |
| raptor | TGAACGTCCTGGGTAAGGTGA | AATGTCGGATATTTGCTCGATGT |
| REPTOR | CCAGCACCACAAATACCTGTAA | CGGTGTCTGCAAATCGGAC |
| REPTOR-BP | ACGCGCAAGGAATGTGGAAA | CAACTACAGCCTTCTCCCGAT |
| Rheb | AGTTCGTGGACTCCTATGACC | ACGATGTAGTCTTGCGACTTTAC |
| rp49 | CTTCATCCGCCACCAGTC | CGACGCACTCTGTTGTCG |
| Tace | TCCTGCACAGCAAATTCAGGG | CACCCGACCCGAATAGAAGC |
| thor/4E-BP | CAGATGCCCGAGGTGTACTC | CATGAAAGCCCGCTCGTAGA |
| unk | CCAGTCGTTTTTACAGCACAAG | CGACGCTGGTTTTGAAAATGC |
| upd3 | AGCCGGAGCGGTAACAAAA | CGAGTAAGATCAGTGACCAGTTC |

## Data availability

The source data of this paper are collected in the following database record: biostudies:S-SCDT-10_1038-S44319-024-00241-3.

## Peer review information

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

## Acknowledgements

We thank Konrad Basler, Joaquim Culi, Rénald Delanoue, Susumu Hirabayashi, Pierre Léopold, Marco Milán, Hugo Stocker, and Aurelio Teleman, for sharing flies and antibodies, Chad Whilding for image analyses, and Marion Barbier for technical help. We acknowledge the imaging facility MRI, a member of the France-BioImaging national infrastructure supported by the French National Research Agency (ANR-10-INBS-04, "Investments for the future"). We acknowledge the Bloomington Drosophila Stock Center (BDSC - NIH P40OD018537), the Vienna Drosophila Resource Center (VDRC), the Developmental Studies Hybridoma Bank, the Montpellier *Drosophila* facility, and FlyBase for providing reagents and tools critical for our research. JF was supported by a PhD fellowship from the French Ministry for Education Research and Technology (MENRT). This project was supported by grants from the "Fondation ARC pour la recherche sur le cancer" #PJA20181207757 and #PJA2023080007002, "GSO-Emergence", and "Agence Nationale de la Recherche #ANR-18-CE14-0041".

## Author contributions

**Miriam Rodríguez-Vázquez**: Conceptualization; Formal analysis; Validation; Investigation; Visualization. **Jennifer Falconi**: Formal analysis; Validation; Investigation. **Lisa Héron-Milhavet**: Validation; Investigation. **Patrice Lassus**: Validation; Investigation. **Charles Géminard**: Formal analysis; Funding acquisition; Validation; Investigation; Visualization; Writing—review and editing. **Alexandre Djiane**: Conceptualization; Formal analysis; Supervision; Funding acquisition; Investigation; Visualization; Writing—original draft; Project administration; Writing—review and editing.

Source data underlying figure panels in this paper may have individual authorship assigned. Where available, figure panel/source data authorship is listed in the following database record: biostudies:S-SCDT-10_1038-S44319-024-00241-3.

## Disclosure and competing interests statement

The authors declare no competing interests.

# Expanded View Figures

**Figure EV1. Fat body atrophy after *Pglym78* knock-down (related to Fig. 1).**

(A) Evaluation of the knock-down efficiency of the different RNAi used monitored by qRT-PCR. Reference gene for normalization: *rp49/RpL32*. Biological replicates $n = 3$. Error bars show the standard error of the mean (sem). One-way ANOVA statistical test, ****$p < 0.0001$. (B) Fat body staining of the indicated genotypes showing nuclei (DAPI, blue) and cell cortex (F-actin, red or white); white bar 50 µm. (C) Quantification of average cell size from images shown in (B), and represented has box plots where the whiskers represent the maxima and minima experimental points, the boxes represent the 25 and 75% percentiles, and where center line is the median. Biological replicates $n = 7$–11. Mann–Whitney test, **$p = 0.0043$. (D) Glycogen content in the adipose tissue after *Pglym78* knock-down compared to control monitored by periodic shift acid staining. White and black bars 200 µm. (E, F) Circulating levels of trehalose in the hemolymph (D left), of glycogen in larvae (D right), and of triglycerides in larvae (TAG; E) after *Pglym78* knock-down in fat body cells compared to controls. Error bars show the standard error of the mean (sem). Biological replicates $n = 3$ to 5. One-way ANOVA statistical test, ns not significant. $p > 0.05$ (E), or Mann–Whitney test, *$p = 0.0260$ (F). (G, H) Quantification of ATP (G) and pyruvate (H) content in dissected fat bodies normalized to total nucleic acid content. Biological triplicates. Error bars show the standard error of the mean (sem). Mann–Whitney test, **$p = 0.0079$ (G) or one-way ANOVA statistical test, *$p = 0.0374$, ns not significant (H). Source data are available online for this figure.

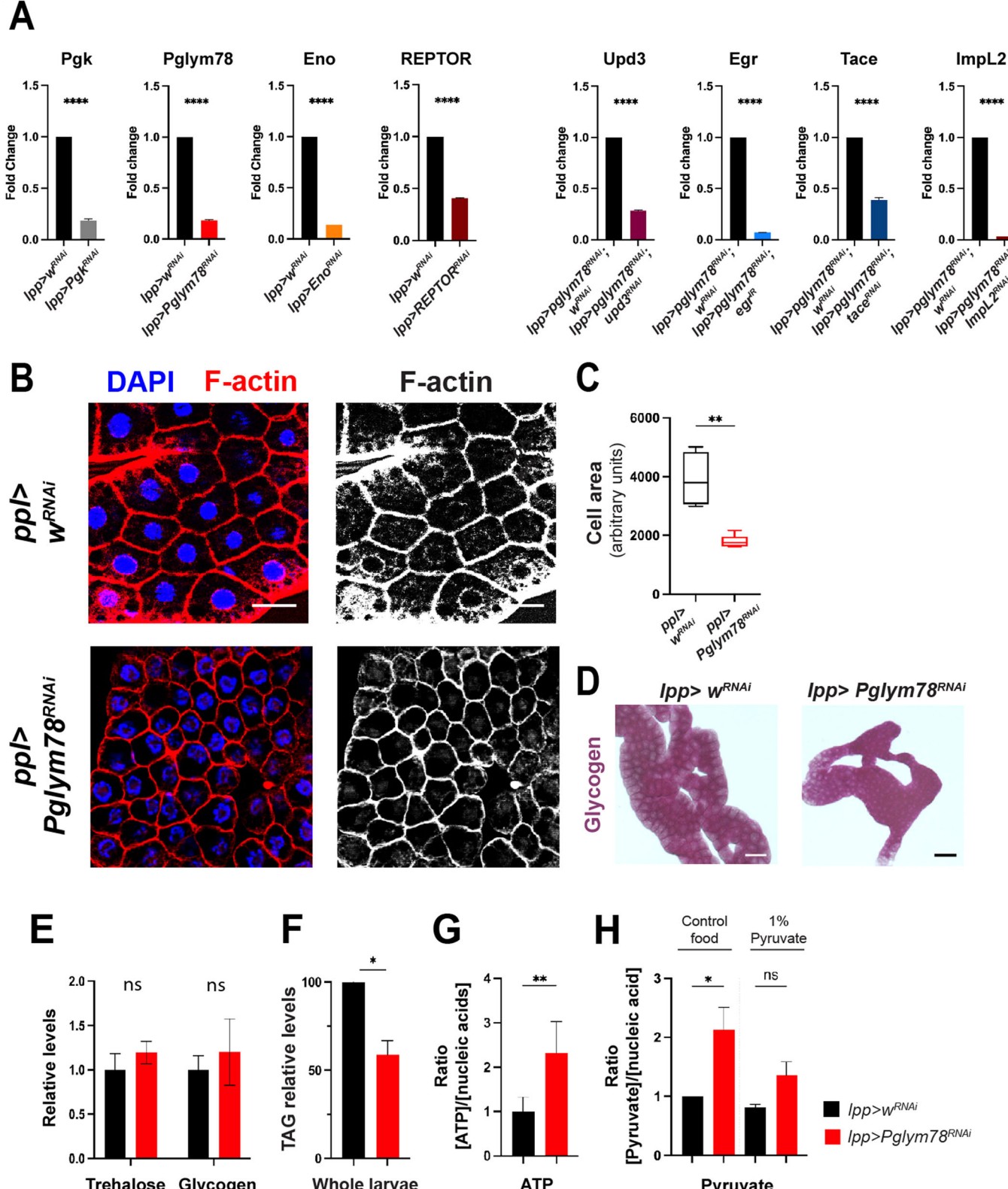

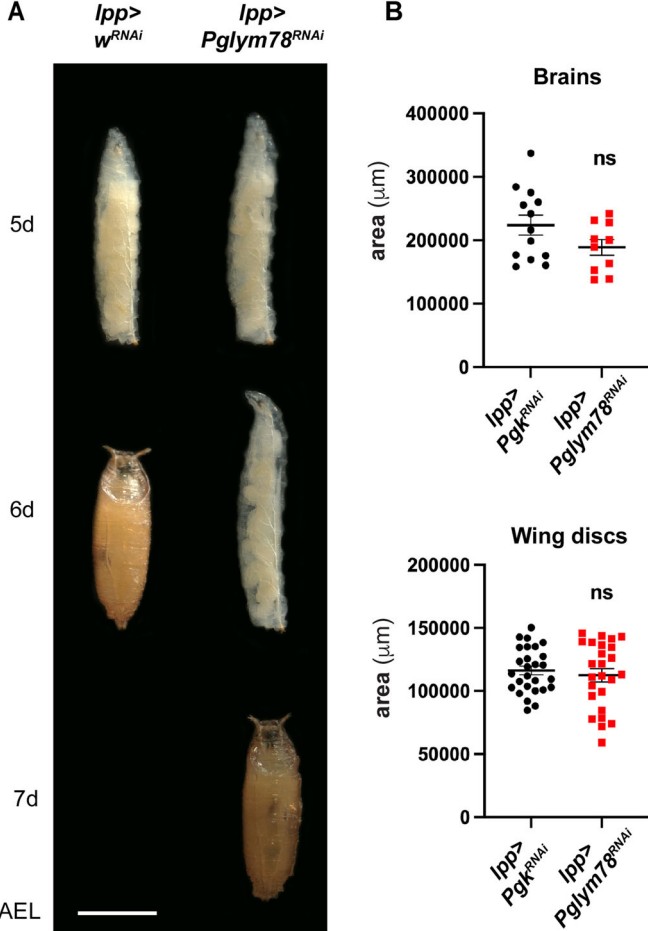

**Figure EV2. Developmental delay after fat body *Pglym78* knock down (related to Fig. 2).**

(A) Images of representative larvae and pupae of *lppGal4> Pglym78-RNAi* and *lppGal4 > w-RNAi* controls at the 5, 6, and 7 days (d) after egg laying (ael); white bar 1 mm. (B) Size of brains and wing disks expressed in arbitrary units (pixels) in *Pglym78-RNAi* animals compared to *Pgk-RNAi* controls at 6 days ael. For brains, $n = 13$ and 10, for disks, $n = 27$ and 24. Mann–Whitney test. ns not significant. Source data are available online for this figure.

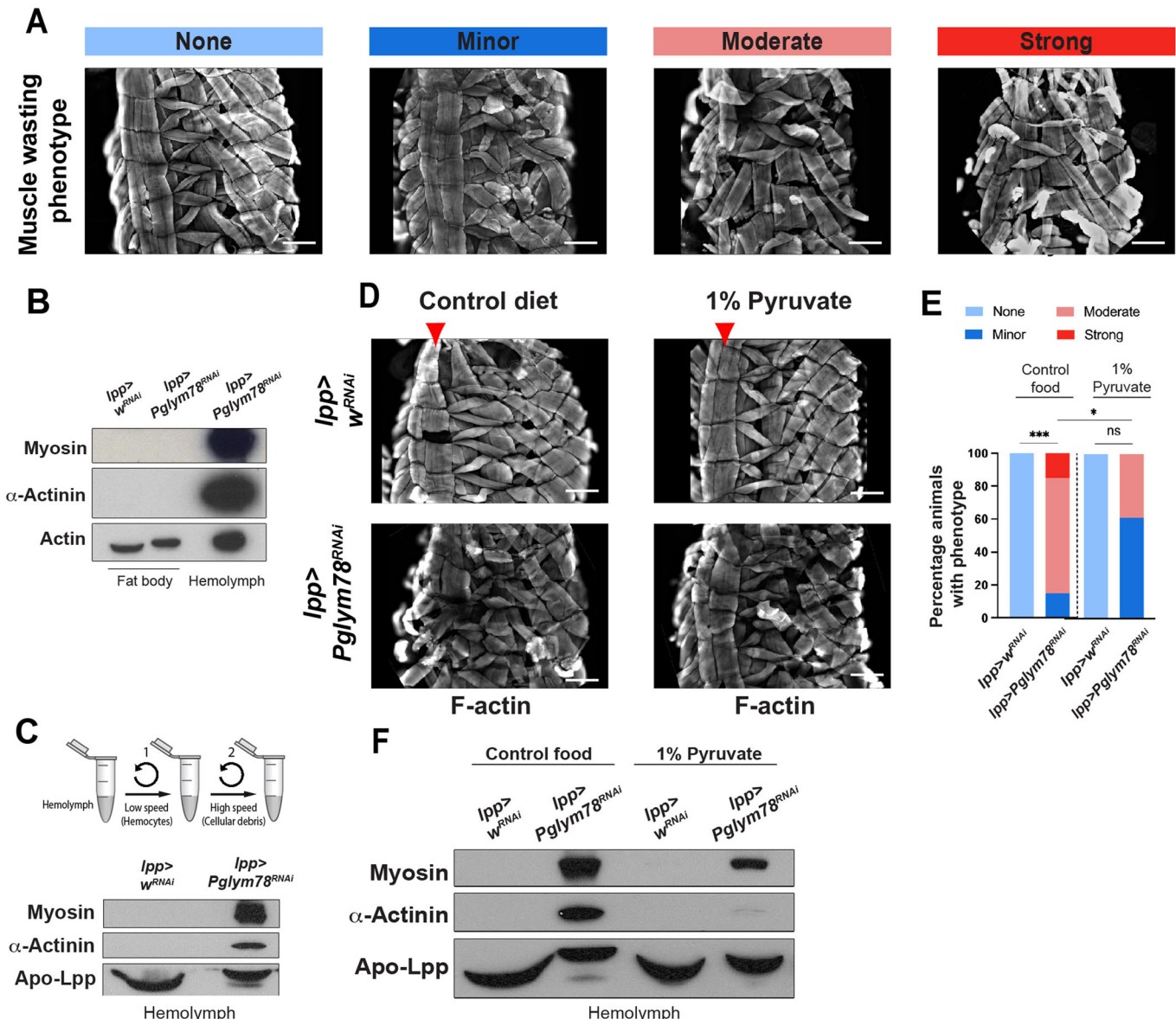

**Figure EV3. Presence of α-actinin in the hemolymph of animals with fat body atrophy (related to Fig. 3).**

(A) Categories used to quantify the muscle disorganization. White bar 200 μm. (B) Western blot of whole protein extracts of fat body tissues and hemolymph of the indicated genotypes monitoring the presence of muscular proteins Myosin and α-Actinin in the fat body. The hemolymph of *lpp-Gal4> Pglym78-RNAi* animals is used as a positive control. Actin is used as a loading control between the two fat body samples. (C) Upper panel: experimental set-up to clear the hemolymph collected from bled larvae: a first centrifugation at low speed to remove circulating cells, and a second at high speed to remove cellular debris. Lower panel: western blot of whole protein extracts of hemolymph (5 μl samples) from *lppGal4> Pglym78-RNAi* larvae after centrifugations and monitoring the presence of muscle proteins Myosin and α-Actinin. Apo-LppII is used as a loading control. (D) Larval body wall muscles of *lppG4> Pglym78-RNAi* animals grown on normal food (left) or food supplemented with pyruvate (right) monitored using F-actin staining (white); white bar 200 μm. The red arrowhead indicates the VL3/VL4 muscles. In all panels anterior is up. (E) Quantification of the muscle defects from (D) and categorized according to severity. For the different genotypes, *n* = 10. Chi-square test, ***p* = 0.0002, **p* = 0.0156, ns not significant. (F) Western blot of whole protein extracts of hemolymph (5 μl samples) from the larvae shown in (D) and monitoring the presence of muscle proteins such as Myosin and α-Actinin. Apo-LppII is used as a loading control. Source data are available online for this figure.

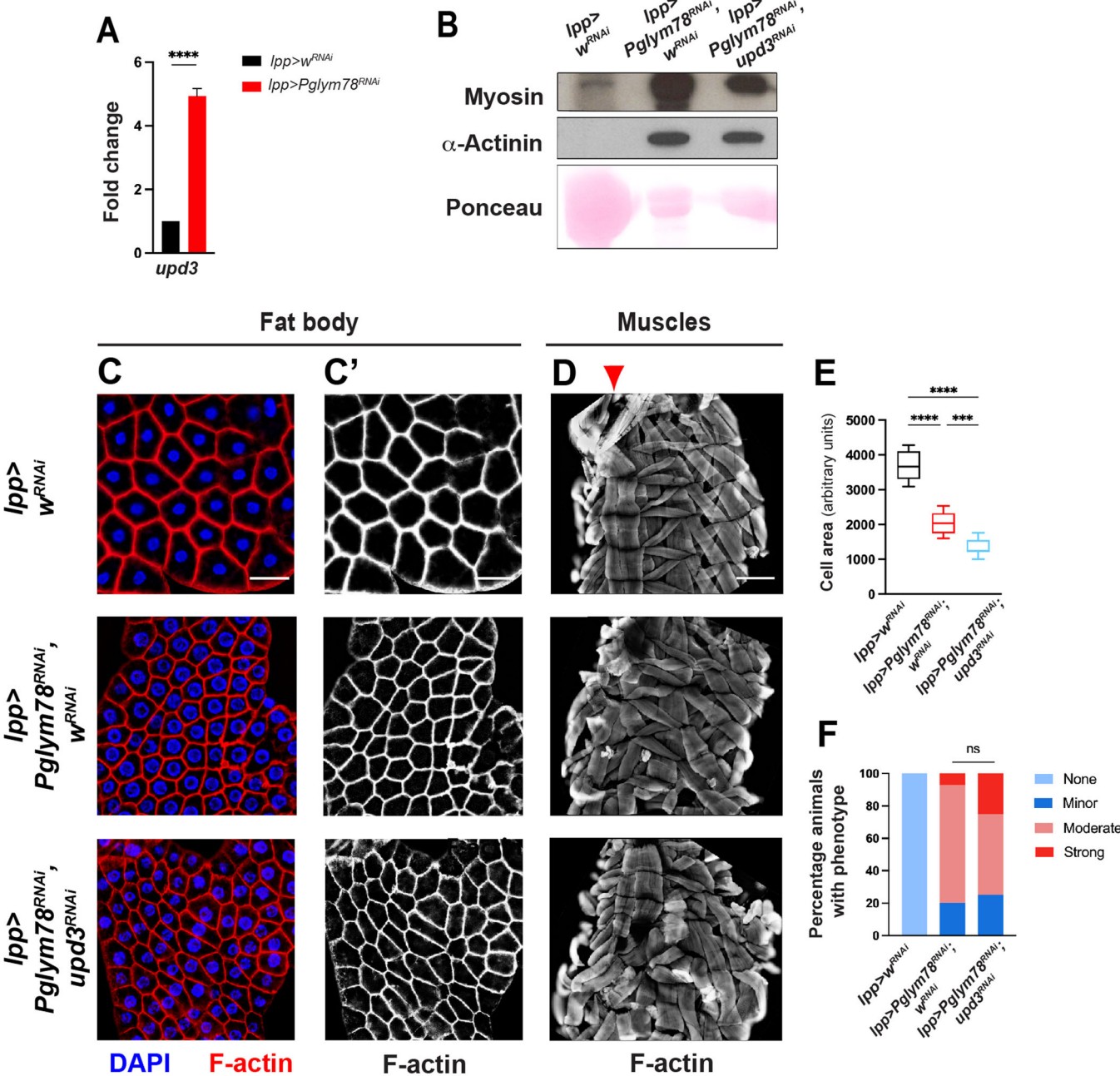

**Figure EV4. Fat body-derived upd3 does not mediate the muscle disorganization triggered by fat body atrophy.**

(A) Expression of the unpaired ligand *upd3* in fat body cells after *Pglym78* knock-down monitored by qRT-PCR. Reference gene for normalization: *rp49/RpL32*. Biological triplicates. Error bars show the standard error of the mean (sem). *T*-test, ****$p < 0.0001$. (B) Western blot of whole protein extracts of hemolymph (5 μl samples) monitoring the presence of muscle proteins such as Myosin and α-Actinin. Ponceau S staining shows the Lsp proteins in the different samples and shows equivalent loading between animals with *Pglym78-RNAi* and animals with combined *Pglym78-RNAi & upd3-RNAi*. Of note, Lsp release is lower in animals with affected fat bodies. (C) Fat body staining of the indicated genotypes showing nuclei (DAPI, blue in C), cell cortex (F-actin, red in B), and white in C'); white bar 50 μm. (D) Larval body wall muscles of the same genotypes as in (C) monitored using F-actin staining (white in D) on dissected fixed larvae; white bar 200 μm. The red arrowhead indicates the VL3/VL4 muscles. In all panels anterior is up. (E) Quantification of average cell size from images shown in (C), and represented has box plots where the whiskers represent the maxima and minima experimental points, the boxes represent the 25 and 75% percentiles, and where center line is the median. Biological replicates $n = 7$–11. One-way ANOVA statistical test, ****$p < 0.0001$, ***$p = 0.0003$. (F) Quantification of the muscle defects from (D) and categorized according to severity. For the different genotypes, $n > 10$. Chi-square test, ns not significant. Source data are available online for this figure.

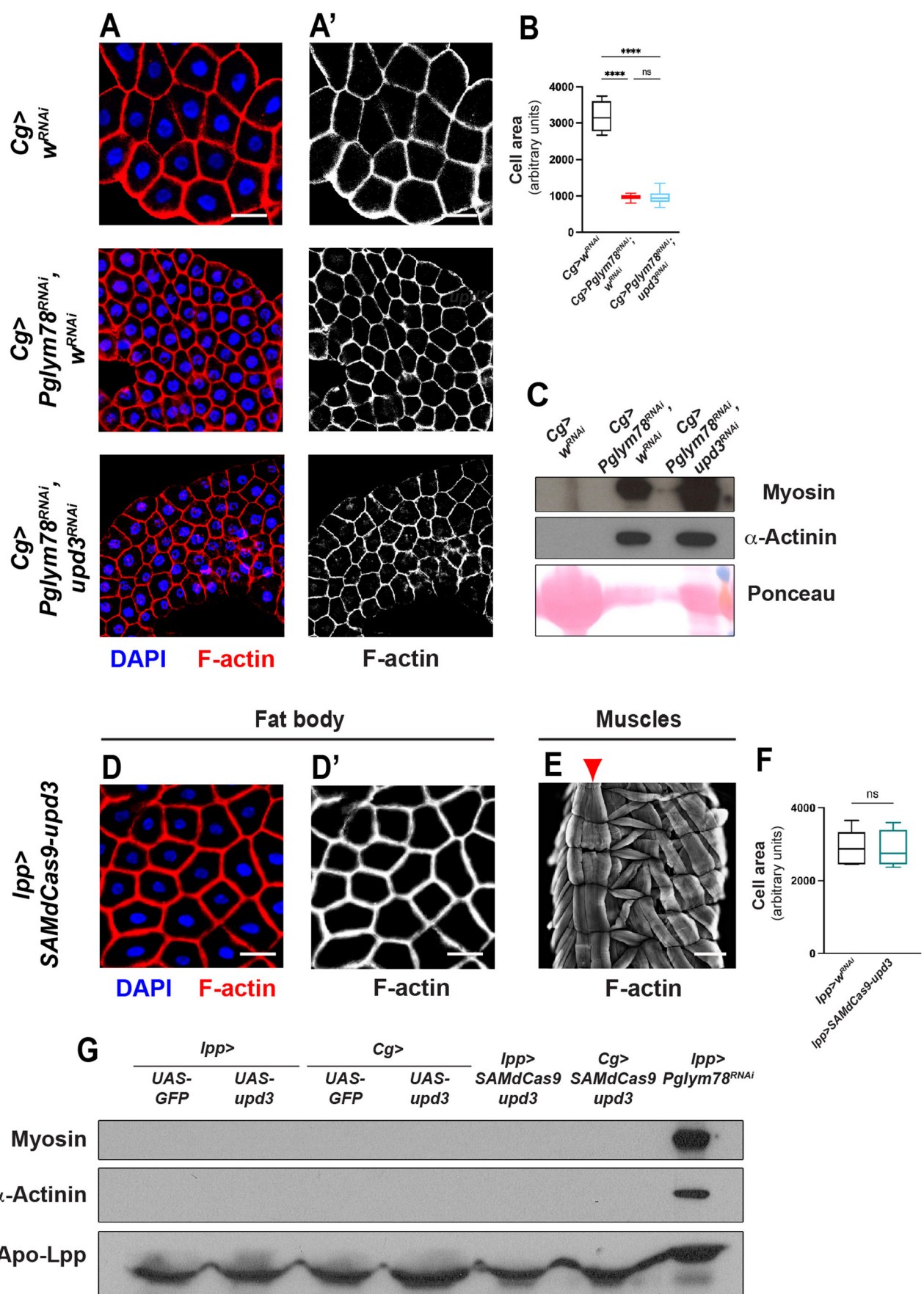

◄ **Figure EV5. Fat body and hemocyte-derived Upd3 is neither required nor sufficient for muscle disorganization.**

(A) Fat body staining after *Pglym78* invalidation in both adipocytes and hemocytes using the Cg-Gal4 driver, and showing nuclei (DAPI, blue in **A**) and cell cortex (F-actin, red in **A** and white in **A'**); white bar 50 µm. (B) Quantification of average cell size from images shown in (**A**), and represented has box plots where the whiskers represent the maxima and minima experimental points, the boxes represent the 25 and 75% percentiles, and where center line is the median. Biological replicates $n = 6$–11. One-way ANOVA statistical test, ****$p < 0.0001$, ns not significant. (C) Western blot of whole protein extracts of hemolymph (5 µl samples) from the larvae shown in (**A**) and monitoring the presence of Myosin and α-Actinin. Ponceau shows the Lsp proteins in the different samples. (D) Fat body staining after upd3 overexpression in adipocytes, and showing nuclei (DAPI, blue in (**D**)) and cell cortex (F-actin, red in (**D**) and white in **D'**); white bar 50 µm. (E) Larval body wall muscles of the same genotypes as in (**D**) monitored using F-actin staining (white in **E**) on dissected fixed larvae; white bar 200 µm. The red arrowhead indicates the VL3/VL4 muscles. Anterior is up. (F) Quantification of average cell size from images shown in (**D**), and represented has box plots where the whiskers represent the maxima and minima experimental points, the boxes represent the 25 and 75% percentiles, and where center line is the median. Biological replicates $n = 8$. Mann–Whitney test, ns not significant. (G) Western blot of whole protein extracts of hemolymph (5 µl samples) from 5d ael larvae overexpressing upd3 either in the adipocytes (*lpp-Gal4*), or in adipocytes and hemocytes (*Cg-Gal4*) and monitoring the presence of Myosin and α-Actinin. The right lane is hemolymph from *lpp-Gal4> Pglym78-RNAi* animals and is used as a positive control for the presence of Myosin and α-Actinin. Apo-LppII is used as a loading control. Source data are available online for this figure.

