## [Peer Review File · EMBO Reports]

Fat body glycolysis defects promote muscle disorganization through egr and ImpL2 signaling.

Miriam Rodríguez-Vázquez, Jennifer Falconi, Lisa Héron-Milhavet, Patrice Lassus, Charles Géminard, and Alexandre Djiane

Corresponding author: Alexandre Djiane (alexandre.djiane@inserm.fr)

Review Timeline:

Submission Date:	28th Oct 23
Editorial Decision:	4th Dec 23
Revision Received:	10th Jun 24
Editorial Decision:	18th Jul 24
Revision Received:	29th Jul 24
Accepted:	9th Aug 24

Editor: Deniz Senyilmaz Tiebe

Transaction Report:

Dear Dr. Djiane,

Thank you for the submission of your research manuscript to our journal, which was now seen by three referees, whose reports are copied below.

My apologies for this unusual delay in getting back to you. It took longer than anticipated to receive the full set of referee reports.

Referees express interest in the proposed metabolic crosstalk between fatbody and muscle through *egr* and *ImpL2*. However, they also raise significant concerns regarding required analyses and missing controls that need to be addressed to consider publication here.

Should you be able to address referee concerns in full, we would like to invite you to submit a revised manuscript. Please revise your manuscript with the understanding that the referee concerns (as in their reports) must be fully addressed and their suggestions taken on board. Please address all referee concerns in a complete point-by-point response. Acceptance of the manuscript will depend on a positive outcome of a second round of review. It is EMBO reports policy to allow a single round of major experimental revision only and acceptance or rejection of the manuscript will therefore depend on the completeness of your responses included in the next, final version of the manuscript.

We realize that it is difficult to revise to a specific deadline. In the interest of protecting the conceptual advance provided by the work, we recommend a revision within 3 months. Please discuss the revision progress ahead of this time with me if you require more time to complete the revisions, or if you have questions or comments regarding the revision (also by video chat).

1. A data availability section providing access to data deposited in public databases is missing (where applicable).
2. Your manuscript contains statistics and error bars based on $n=2$. Please use scatter plots in these cases.

You can submit the revision either as a Scientific Report or as a Research Article. For Scientific Reports, the revised manuscript can contain up to 5 main figures and 5 Expanded View figures, and it should not exceed 27000 characters. If the revision leads to a manuscript with more than 5 main figures it will be published as a Research Article. In this case the Results and Discussion section should be separate. If a Scientific Report is submitted, these sections have to be combined. This will help to shorten the manuscript text by eliminating some redundancy that is inevitable when discussing the same experiments twice. In either case, all materials and methods should be included in the main manuscript file.

3) We replaced Supplementary Information with Expanded View (EV) Figures and Tables that are collapsible/expandable online. A maximum of 5 EV Figures can be typeset. EV Figures should be cited as 'Figure EV1, Figure EV2' etc... in the text and their respective legends should be included in the main text after the legends of regular figures.

4) a .docx formatted letter INCLUDING the reviewers' reports and your detailed point-by-point responses to their comments. As part of the EMBO publication's Transparent Editorial Process, EMBO reports publishes online a Review Process File (RPF) to accompany accepted manuscripts. This File will be published in conjunction with your paper and will include the referee reports, your point-by-point response and all pertinent correspondence relating to the manuscript.

<https://www.embopress.org/page/journal/14693178/authorguide#transparentprocess>

5) a complete author checklist, which you can download from our author guidelines

<https://www.embopress.org/page/journal/14693178/authorguide>. Please insert information in the checklist that is also reflected in the manuscript. The completed author checklist will also be part of the RPF.

6) Please note that all corresponding authors are required to supply an ORCID ID for their name upon submission of a revised manuscript (<<https://orcid.org/>>). Please find instructions on how to link your ORCID ID to your account in our manuscript tracking system in our Author guidelines

<<https://www.embopress.org/page/journal/14693178/authorguide#authorshipguidelines>>

7) Before submitting your revision, primary datasets produced in this study need to be deposited in an appropriate public database (see <https://www.embopress.org/page/journal/14693178/authorguide#datadeposition>). Please remember to provide a reviewer password if the datasets are not yet public. The accession numbers and database should be listed in a formal "Data Availability" section placed after Materials & Method (see also

<https://www.embopress.org/page/journal/14693178/authorguide#datadeposition>). Please note that the Data Availability Section is restricted to new primary data that are part of this study. * Note - All links should resolve to a page where the data can be accessed. *

Additional information on source data and instruction on how to label the files are available:

<https://www.embopress.org/page/journal/14693178/authorguide#sourcedata>

9) Our journal encourages inclusion of *data citations in the reference list* to directly cite datasets that were re-used and obtained from public databases. Data citations in the article text are distinct from normal bibliographical citations and should directly link to the database records from which the data can be accessed. In the main text, data citations are formatted as follows: "Data ref: Smith et al, 2001" or "Data ref: NCBI Sequence Read Archive PRJNA342805, 2017". In the Reference list, data citations must be labeled with "[DATASET]". A data reference must provide the database name, accession number/identifiers and a resolvable link to the landing page from which the data can be accessed at the end of the reference. Further instructions are available at <http://www.embopress.org/page/journal/14693178/authorguide#referencesformat>

10) Regarding data quantification (see Figure Legends:

<https://www.embopress.org/page/journal/14693178/authorguide#figureformat>)

12) Please also note our reference format:

I look forward to seeing a revised version of your manuscript when it is ready. Please let me know if you have questions or comments regarding the revision.

Kind regards,

Deniz Senyilmaz Tiebe

Deniz Senyilmaz Tiebe, PhD
Scientific Editor
EMBO Reports

Referee #1:

This manuscript by Rodriguez-Vasquez and colleagues examines how disruption of glycolysis within the larval fat body alters muscle integrity. Using an RNAi based approach, the authors demonstrate that depletion of the glycolytic enzymes Enolase and Pgym78 within the larval fat body not only induce cell autonomous defects but also results in nonautonomous degradation of muscle. Notably, authors observe that loss of Pgym78 within the fat body results in muscle-specific proteins being released into the hemolymph. Finally, the authors conduct a series of experiments to determine the molecular signal that is inducing fat body wasting. In this regard, they made four major observations: (i) disruption of glycolysis in the fat body results in REPTOR activation, (ii) Upd3 is not essential for the signal, and changes in both (iii) TNF α and (iv) insulin signaling are required for the phenotype.

Overall, I found this to be an exciting story - some of the strongest evidence in the *Drosophila* system that I can think of linking carbohydrate metabolism with endocrine signaling. The study will also be of wide interest to the developmental metabolism community. However, I would ask that the authors address the following issues with the manuscript:

1. The authors should use qRT-PCR to characterize the knockdown efficiency of the RNAi transgenes. In addition, does knockdown of individual glycolytic enzymes induce a lethal phenotype?
2. Related to point 1, the manuscript focuses, in part, on disruption of TAG accumulation, however, this phenotype is never quantified. While I appreciate the use of Bodipy in describing the phenotype, a quantitative measurement of TAG accumulation is still warranted.
3. One of the more interesting findings in the paper is that feeding pyruvate can partially rescue lipid droplet content. However, this observation assumes that pyruvate levels are decreased in the RNAi treated animals and increased by feeding. I'd request that the authors confirm these assumptions by measuring pyruvate levels in treated animals.
4. In Figure 4B, the authors quantify pS6k but do not measure S6k. Based on this data, the change in pS6k abundance could stem from either changes in phosphorylation or total S6k abundance. The authors should determine if S6k abundance is altered in the RNAi background.

A few additional comments:

1. Apologies if I missed this, but Figure 4F doesn't seem to be discussed in the text. Please discuss as appropriate.
2. "knockdown of the upper glycolysis enzyme Pgk" - I'm not sure what the authors mean by "upper". Please clarify or remove from the text.

Referee #2:

Rodríguez-Vázquez et al. describe that impairment of glycolysis in the larval fat body disrupts the organization of body wall

muscles. Using a candidate-based approach, the authors identified a few players that work in the fat body and are required for the disorganization of body wall muscles. The paper provides an interesting perspective on how impairment of glycolysis in the metabolic hub impacts a remote tissue involving secreted factors implicated in tumor-induced tissue wasting. Moreover, the study discovers a potential serological marker of muscle degeneration. The study is of general interest and would be suitable for publication in EMBO reports if the authors address the following comments:

Major comments:

- 1) Quantification of the muscle disorganization would be very helpful for precisely assessing the phenotype. It is not clear how much rescue is observed based on the representative images of body wall muscles. Additionally, the quantification of adipocyte size is missing in some figures.
- 2) A few possible explanations regarding fat body atrophy induced by perturbation of glycolysis are provided in the discussion. In particular, the authors speculated that pglum78 RNAi would lead to a reduction in ATP levels, leading to the activation of AMPK. Nevertheless, fatty acid oxidation might be sufficient to produce enough ATP. On the other hand, glycolysis also produces acetyl-CoA, which is a precursor for fatty acid synthesis. Moreover, glycolysis is tied to the pentose phosphate pathway, which produces NADPH for fatty acid synthesis and nucleotide synthesis. Thus, an impairment of glycolysis can directly impact fat storage in the adipocytes. I suggest the authors cover this alternative possibility in the discussion. Measuring ATP levels in a tissue is straightforward and should help us to understand the basis of fat body atrophy.
- 3) On page 7, "Taken together, these results show that upon glycolysis shut down,.....by increasing REPTOR activity, leading...." The authors concluded that REPTOR activity is increased in the pglum87 RNAi fat body. Nevertheless, REPTOR activity has not been measured, and mRNA levels suggest the opposite. Moreover, it is not clear how REPTOR is linked to fat body atrophy and the induction of muscle disorganization. So, I suggest revising the conclusion to reflect the data presented in the manuscript.
- 4) The experiments indicate that *egr* in the fat body is required for muscle disorganization. It will significantly strengthen the story if the authors show that TNF signaling is increased in the muscle. In particular, TNF signaling leads to activation of JNK and Dronc, which induces apoptosis. It is possible that TNF-induced apoptosis in the muscles might cause muscle disorganization. JNK activity can be assessed by checking phospho-JNK levels and using *puc-lacZ*. An antibody for detecting human cleaved caspase-3 can be used for checking the Dronc activity. A significant increase in *ImpL2* induces tissue atrophy, but it may not be sufficient to induce tissue degeneration. Thus, the cooperation of *egr* and *ImpL2* might account for muscle disorganization as it was discussed. Testing whether simultaneous overexpression of *egr* and *ImpL2* may provide new insight into their roles in tissue degeneration.
- 5) The second paragraph in the discussion (page 11) starting with "Egr/TNF-alpha" appears too speculative. Egr mRNA levels are reduced. Additionally, it is not shown if TNF signaling is activated in the fat body or body wall muscles.
- 6) In the third paragraph of the discussion (page 11), "Alternatively, ageing". It is not clear how larval muscle degeneration phenotype is compared to muscle aging, given that their physiological contexts are completely different. Those sentences can be removed or should provide additional arguments.
- 7) On page 12, "while the exact..... secreted by the fat body". No data is shown to support that REPTOR controls what nutrients are stored or secreted by the fat body. The sentence should be deleted or appropriately revised.
- 8) It is not clearly described how glycogen and trehalose are measured and how they are normalized. Are they normalized to protein levels (in hemolymph)? Is concentration measured or the total amount?
- 9) Did the authors test if RNAi lines used in the study efficiently knockdown the target mRNAs?
- 10) Based on the description in the method section, the data presented in Figure 4A show fold changes. How is the fold change calculated? If fold changes are shown, the y-axis label needs to be revised. It could be just an error, but the values for *lpp>wRNAi* are not 1.0. Is it because the values are normalized in a peculiar way? Would it be more appropriate to have the values for *lpp>wRNAi* normalized to be 1.0?
- 11) The study provides an interesting possibility of using hemolymph alpha-actinin as a molecular marker for muscle degeneration. It will make the finding very exciting to those who study cachexia using *Drosophila* tumor models if the authors show that alpha-actinin can be also detected in the hemolymph of larvae or adult flies bearing tumors inducing muscle degeneration.

Referee #3:

In this study, Rodriguez-Vazquez and colleagues investigate an inter-tissue communication mechanism between the fat body and muscle in *Drosophila* larvae. The authors utilize in vivo *Drosophila* genetic models, attenuating key glycolytic enzymes in the fat body, to induce fat body atrophy, which further leads to muscle disorganization (wasting) through a potential systemic mechanism - implicating TOR signaling in the fat body, as well as Eiger (TNF) and *ImpL2*, as part of a communication signaling network between fat body and muscle in *Drosophila* larvae.

Overall, the manuscript attempts to address an interesting question related to growth coordination during development, potentially linking key metabolic cycles in nutrient storage tissues to proper growth of distal/peripheral tissues through distinct signaling mechanism.

However, I have some major conceptual and experimental concerns with the manuscript, which include a general lack of clarity related to the biological relevance of the findings (associated with both the genetic model of glycolysis inhibition and the tissue

communication axis) and a lack of sufficient experimental justification for many of the conclusions related to mechanism. Below are some major concerns:

1. The authors induce fat body atrophy through inhibiting key glycolytic enzymes, which drives autonomous lipid metabolic dysfunction and at least one sign of systemic muscle wasting. These genetic models lead to either very mild or severe developmental phenotypes. However, it's unclear if glycolysis is an important feature of this inter-organ communication, or just an artificial model to severely limit fat body function. This diminishes the overall biological relevance of the larger phenotypic findings and mechanisms. Just as a few examples:

While the authors supplement pyruvate in the fly food and show a rescue of fat body atrophy in *LppGal4>Pglym78* or *Eno RNAi* larvae, they never check if pyruvate supplementation impacts developmental phenotypes, muscle wasting, or changes in gene expression (used mainly to imply mechanism). The authors never measure pyruvate or other metabolites in their various genetic models to explore how glycolysis (or other impacted metabolic cycles) are impacted - maybe important since carbohydrate storage (and circulation) was not impacted?. Etc.

2. Related to the over-arching mechanisms and the proposed model in Figure 7: in general, there is an over-reliance on genetic interactions and a lack of supplemental experimental analysis that would support the mechanistic conclusions provided in the manuscript. For example, while an increase in REPTOR activity was never explicitly shown, the authors conclude that 'these results show that upon glycolysis shutdown, adipocyte atrophy is mediated, at least in part, by increased REPTOR activity, leading to distant muscle disorganization'. What is the connection between glycolytic inhibition and the TOR pathway?, is it direct or indirect?, would this be true of any genetic model which induced fat body atrophy, and not specific to glycolysis? These same questions can be asked of the mechanistic link between glycolytic-induced fat body atrophy and changes in Eiger or ImpL2. Thus, it is impossible to determine the directness or indirectness of some of these potential mechanisms, whether it be the link between glycolysis and TOR/REPTOR activity, the link between glycolysis/TOR and Eiger or ImpL2 secretion/function, etc. In other the words, the arrows in the proposed model, as suggested, are inconclusive as it relates to the interconnectedness of these mechanism(s), making the study very correlative and potentially (without further experimental analysis) a bit artificial. I also find it odd that an entire main Figure was used to highlight that *Upd3* is likely not inducing muscle wasting (negative result), when the next Figure does highlight a potential secreted factor that could be driving this inter-organ communication.

3. In general, more data (controls and otherwise) needs to be provided to justify some of the findings. Just as a few examples: The authors should include additional measurements for muscle disorganization/wasting (especially linked to function) and specific mechanisms underlying muscle wasting should be discussed. Why is there such a massive difference in developmental phenotypes when attenuating *Eno* vs *Pglym78*, and are these potential different models of metabolic deficiency and thus influencing different types of mechanisms? Etc.

Referee #1:

This manuscript by Rodriguez-Vasquez and colleagues examines how disruption of glycolysis within the larval fat body alters muscle integrity. Using an RNAi based approach, the authors demonstrate that depletion of the glycolytic enzymes Enolase and Pglym78 within the larval fat body not only induce cell autonomous defects but also results in nonautonomous degradation of muscle. Notably, authors observe that loss of Pglym78 within the fat body results in muscle-specific proteins being released into the hemolymph. Finally, the authors conduct a series of experiments to determine the molecular signal that is inducing fat body wasting. In this regard, they made four major observations: (i) disruption of glycolysis in the fat body results in REPTOR activation, (ii) Upd3 is not essential for the signal, and changes in both (iii) TNF α and (iv) insulin signaling are required for the phenotype.

Overall, I found this to be an exciting story - some of the strongest evidence in the *Drosophila* system that I can think of linking carbohydrate metabolism with endocrine signaling. The study will also be of wide interest to the developmental metabolism community. However, I would ask that the authors address the following issues with the manuscript:

1. The authors should use qRT-PCR to characterize the knockdown efficiency of the RNAi transgenes. In addition, does knockdown of individual glycolytic enzymes induce a lethal phenotype?

We have included data on the RNAi knock-down of the different lines used in the Fig. EV1A. Of note, all of the glycolysis enzymes RNAi used were of similar strength with a knock-down efficiency greater than 80%. This has now been added in the text on page 5: "Importantly, the knock-down efficiencies of the different RNAi used were very similar and reached more than 80% for *Pgk*, *Pglym78*, and *Eno*, suggesting that the differences observed between *Pgk* and *Pglym78* were not caused by differences in RNA knock-down (Fig. EV1A)".

For the functional tests, the RNAi efficiencies were between 60% and 95%. Nevertheless we observed modifications of the phenotypes for *REPTOR*, *egr*, *Tace*, and *Impl2* showing that even though partial, the knock-downs were sufficient to modify the *Pglm78-RNAi* phenotypes. For the Upd ligands, we removed the Upd1 and Upd2 results. Indeed, we felt this was providing too much negative results, and Upd1 and Upd2 have not been linked to muscle atrophy. In our muscle disorganisation modifier candidate approach we thus focus the text and results on Upd3 which has been linked to muscle atrophy in adult cachexia models. The knock-down of *upd3*, even though relatively strong (70%), was not able to restore "normal" levels similar to those found in a wild-type fat body, and we cannot exclude that the lack of modifications observed after knocking down *upd3* could be explained by its residual extra expression.

See also comment #9 from Referee #2.

Knock-down of the glycolytic enzymes Pglym78 or Eno with *lpp-Gal4* or *ppl-Gal4* do give a lethal phenotype: *Pglym78-RNAi* die as pupae, while *Eno-RNAi* die as larvae. This is shown in Fig. 2 and explained in the text. For *Pgk-RNAi*, pupariation is delayed, but adults do emerge.

2. Related to point 1, the manuscript focuses, in part, on disruption of TAG accumulation, however, this phenotype is never quantified. While I appreciate the use of Bodipy in describing the phenotype, a quantitative measurement of TAG accumulation is still warranted.

We have now added TAG measurements as part of the phenotypic description of *Pgk* and *Pglym* RNAi animals in the text (page 5), and as new panels in Fig. 1E and Fig. EV1D. Consistent with BODIPY staining and quantification there is less TAG in *Pglym78-RNAi* fat bodies. We performed these measurements on dissected fat bodies (Fig. 1E) and on whole larvae (Fig. S1D), giving similar reduction in TAG. For whole larvae, we normalised by larvae numbers. Indeed, most TAG reserves are in the fat body and we reasoned that whole larvae measures would reflect the state of reserves in Fat bodies. But we also performed TAG measurements on dissected fat bodies. For dissected tissues, we had to normalise by another measurement. This represent a challenging task for severely atrophied tissues where almost everything is affected. The fat body represents in *Drosophila* larvae a major organ for protein reserves, which are very likely highly impacted in *Pglym78-RNAi*. Indeed, Lsp proteins, produced by the fat body and released in the haemolymph, are dramatically reduced upon *Pglym78* RNAi. We thus normalised TAG (and the other metabolites monitored) to total nucleic acid content, which could represent a proxy to cell numbers and is likely amongst the least affected compound.

3. One of the more interesting findings in the paper is that feeding pyruvate can partially rescue lipid droplet content. However, this observation assumes that pyruvate levels are decreased in the RNAi

treated animals and increased by feeding. I'd request that the authors confirm these assumptions by measuring pyruvate levels in treated animals.

We thank the referee for this insight which helps clarify the results and the text. Beside the problem of normalisation mentioned above, measuring steady state levels of metabolites, while informative, can be difficult to interpret. We measured pyruvate levels in the atrophied *Pgkym78-RNAi* fat body, and show that steady state levels normalised to total nucleic acid content actually increase following glycolysis impairment (Fig. EV1). We interpret this result as a sign that upon *Pgkym78-RNAi*, cells rewired their metabolism and stopped using pyruvate to generate energy. Pyruvate, the final product of glycolysis is no longer used and accumulates even if produced at a reduced rate due to the *Pgkym78-RNAi*. Indeed, when we supplemented the food with 1% pyruvate, the steady state levels of pyruvate go back to normal, indicative that pyruvate is now again used as a source for Ac-CoA and energy. This is now shown in Fig. EV1.

Since TAG levels were decreased and that lipid droplets shrank (BODIPY), the metabolic rewiring is likely a shift to β -oxidation of lipids as source of Ac-CoA which can then fuel the TCA cycle.

This shift from glycolysis to β -oxidation as source of energy is now discussed in the results and in the discussion on pages 6 and 12.

See also comment #2 from Referee #2.

4. In Figure 4B, the authors quantify pS6k but do not measure S6k. Based on this data, the change in pS6k abundance could stem from either changes in phosphorylation or total S6k abundance. The authors should determine if S6k abundance is altered in the RNAi background.

We now show that total S6k levels, which remained constant between conditions, as loading control (Figure 4B). The ratio pS6k / total S6k is thus lower in *Pgkym78-RNAi* compared to similarly aged controls, supporting lower mTOR activity.

A few additional comments:

1. Apologies if I missed this, but Figure 4F doesn't seem to be discussed in the text. Please discuss as appropriate.

Figure 4F, which corresponds now to Figure 4I in the reviewed figures, was indeed discussed in the text on page 9 under the paragraph "***The mTOR effector REPTOR is required for fat body atrophy and distant muscle wasting***" of the Results section.

"Similarly, the intensity of F-actin staining in muscles, and in particular in the VL3 and VL4 muscles were restored, and α -Actinin protein levels in the circulating hemolymph were reduced (Fig. 4I)."

2. "knockdown of the upper glycolysis enzyme Pgk" - I'm not sure what the authors mean by "upper". Please clarify or remove from the text.

Yes, we meant the enzyme above *Pgkym78* in the glycolysis. This has now been changed in page 5. The original sentence "The knock down of the upper glycolysis enzyme Pgk had a weaker effect under these conditions (Fig. 1B&C)." has been changed to "The knock down of Phosphoglycerate kinase (Pgk), the glycolytic enzyme just above *Pgkym78*, had a weaker effect under these conditions (Fig. 1B&C)."

Referee #2:

Rodríguez-Vázquez et al. describe that impairment of glycolysis in the larval fat body disrupts the organization of body wall muscles. Using a candidate-based approach, the authors identified a few players that work in the fat body and are required for the disorganization of body wall muscles. The paper provides an interesting perspective on how impairment of glycolysis in the metabolic hub impacts a remote tissue involving secreted factors implicated in tumor-induced tissue wasting. Moreover, the study discovers a potential serological marker of muscle degeneration. The study is of general interest and would be suitable for publication in EMBO reports if the authors address the following comments:

Major comments:

1) Quantification of the muscle disorganization would be very helpful for precisely assessing the phenotype. It is not clear how much rescue is observed based on the representative images of body wall muscles. Additionally, the quantification of adipocyte size is missing in some figures.

We provide the quantification of body wall muscle disorganisation phenotype in the figures and legends. Quantification methods have been added in the Material and Methods section and are based on previously published method in Newton et al. Nat Commun 2020 (doi:10.1038/s41467-020-18502-9). Briefly, muscle disruption was categorised into: none, minor, medium, strong disruption. Statistics on frequency changes compared to the expected frequencies in *white-RNAi* or *Pglym78-RNAi* controls were assessed using the chi squared statistical test. We report that the frequency of medium and strongly affected muscles increases in *Pglym78* or *Eno RNAi* compared to *white-RNAi* controls, and that this increase is suppressed when deleting *egr*, *Tace*, or *ImpL2*.

We now provide the quantification of adipocyte cell size in all experiments.

2) A few possible explanations regarding fat body atrophy induced by perturbation of glycolysis are provided in the discussion. In particular, the authors speculated that *pglym78 RNAi* would lead to a reduction in ATP levels, leading to the activation of AMPK. Nevertheless, fatty acid oxidation might be sufficient to produce enough ATP. On the other hand, glycolysis also produces acetyl-CoA, which is a precursor for fatty acid synthesis. Moreover, glycolysis is tied to the pentose phosphate pathway, which produces NADPH for fatty acid synthesis and nucleotide synthesis. Thus, an impairment of glycolysis can directly impact fat storage in the adipocytes. I suggest the authors cover this alternative possibility in the discussion. Measuring ATP levels in a tissue is straightforward and should help us to understand the basis of fat body atrophy.

We thank the referee for this insight. The links between glycolysis and lipid reserves have indeed been the subject of intense research, including in *Drosophila* fat body adipocytes. The primary goal of the study here was not the dissection of the biochemical links between glycolysis and fat body atrophy, but rather to understand the global effects of altering the metabolically active fat body tissue. Following the referee suggestions, we now include in the discussion a paragraph stressing the potential links between glycolysis and lipid reserves, including a balance between lipolysis and de-novo lipogenesis on pages 5 and 12. Our results measuring pyruvate and BODIPY/ TAG, point toward a major metabolic switch in *Pglym78-RNAi* where upon glycolysis blockade, cells adapt and resort to β -oxidation as a source of energy, compatible with a strong depletion of lipid reserves.

See comment #3 from Referee #1

We have measured steady state ATP levels in fat body cells after *Pglym78* invalidation, which proved less straightforward than anticipated, largely in part due to normalisation issues. Normalisation for severely atrophied tissues, where almost everything is affected, represents a challenge. The fat body represents in *Drosophila* larvae a major organ for protein reserves, which are very likely highly impacted in *Pglym78-RNAi*. We thus normalised metabolites measurements to total nucleic acid content, which could represent a proxy to cell numbers and is likely amongst the least affected compound.

We observed a strong increase in ATP levels normalised to nucleic acid (Fig. EV1). This result thus shows that after *Pglym78-RNAi* and blockage of glycolysis, cells adapt and maintain their ATP levels. Steady state levels of ATP actually increase, either due to lesser consumption or increased production. Reports in cancer cells have shown that chemotherapy-resistant cancer cell lines which switch from glycolysis to β -oxidation have higher intracellular ATP levels, but the exact mechanisms underlying the increased steady state ATP levels observed in our experimental set up remain to be identified.

We mentioned AMPK activation as a potential signal for the inhibition of the mTOR pathway upon *Pglym78-RNAi* in the discussion. However, opposite to our candid initial hypothesis, we now show that ATP levels were increased when glycolysis was blocked. Even though we did not measure the ATP/ADP ratio, we show that activated AMPK (pAMPK) levels were reduced after *Pglym78* invalidation. Because the total AMPK antibody does not work on the *Drosophila* protein we could not compare to total AMPK levels and normalised compared to Histone H3 (Figure 4B). We acknowledge that this is not ideal, but our results show that for S6K, *Pglym78* invalidation did not affect total enzyme levels. Together with increased ATP levels, it is thus unlikely that the mTOR inhibition is mediated by increased AMPK activity. We have thus updated this part of the discussion.

3) On page 7, "Taken together, these results show that upon glycolysis shut down,.....by increasing REPTOR activity, leading...." The authors concluded that REPTOR activity is increased in the *pglum87* RNAi fat body. Nevertheless, REPTOR activity has not been measured, and mRNA levels suggest the opposite. Moreover, it is not clear how REPTOR is linked to fat body atrophy and the induction of muscle disorganization. So, I suggest revising the conclusion to reflect the data presented in the manuscript.

We thank the referee for this insight. We agree with the referee, that REPTOR activity was not measured in the original version of the manuscript. Even though the target genes of REPTOR in larval fat body cells have not been formally identified, we have used two commonly reported REPTOR transcriptional targets as REPTOR activity reporters: *4E-BP/thor* and *unk*. *4E-BP/thor* is also controlled by other pathways including Insulin/FOXO signalling, but *unk* represents a relatively specific target gene of REPTOR.

We show here that mTOR pathway activity is lower in *Pglym78-RNAi* fat body cells. However, *4E-BP* and *unk* transcripts levels were down, suggesting that REPTOR activity is down-regulated in *Pglym78-RNAi* cells. We propose that this might be due to lower levels of REPTOR-BP, whose transcripts levels are low in *Pglym78-RNAi* cells. We propose in the discussion that this low level of REPTOR/REPTOR-BP might represent an adaptation of cells to prevent excessive catabolism in conditions where cells have already almost used up their lipid reserves. These results have been added in Fig 4C.

However, while REPTOR activity was diminished in *Pglym78-RNAi* cells, it was still active since lowering REPTOR levels by RNAi further diminished *4E-BP* and *unk* expression levels (Fig. 4D). Removing REPTOR also efficiently suppressed fat body atrophy suggesting that either REPTOR activity was critical at earlier stages for atrophy, or that the remaining REPTOR activity participates in the fat body atrophy.

We have included all these new data, suggesting that the link between mTOR and REPTOR might be more complicated in the context of *Pglym78-RNAi*, and thus moderated the title of the paragraph to "The mTOR effector REPTOR is required for fat body atrophy and distant muscle wasting"

4) The experiments indicate that *egr* in the fat body is required for muscle disorganization. It will significantly strengthen the story if the authors show that TNF signaling is increased in the muscle. In particular, TNF signaling leads to activation of JNK and Dronc, which induces apoptosis. It is possible that TNF-induced apoptosis in the muscles might cause muscle disorganization. JNK activity can be assessed by checking phospho-JNK levels and using *puc-lacZ*. An antibody for detecting human cleaved caspase-3 can be used for checking the Dronc activity. A significant increase in *ImpL2* induces tissue atrophy, but it may not be sufficient to induce tissue degeneration. Thus, the cooperation of *egr* and *ImpL2* might account for muscle disorganization as it was discussed. Testing whether simultaneous overexpression of *egr* and *ImpL2* may provide new insight into their roles in tissue degeneration.

We thank the reviewer for these suggestions.

i) We remained cautious about the direct or indirect mechanism of *egr* on muscles in the discussion, but the referee asks here to perform experiments that could provide hints towards a direct role. We have generated double balanced lines to be able to test whether disorganised muscles activate JNK activity. In muscles we did not observe any significant changes (activation) in JNK activity using the reporters *puc-LacZ* and *TRE-DsRed*. Similarly, we could not detect any significant increase in caspase activity through Dcp-1 staining. In our hands, the original batch of anti-cleaved Caspase-3 could indeed cross-react with *Drosophila* Caspases, but this was lost in subsequent batches of the antibody. We thus cannot show any hints that *egr* could act directly on the muscles, even though it remains possible that the effects might be too subtle to probe, or that *egr* on muscles acts through alternative pathways.

Please find below representative images of the *puc-LacZ* reporters stain in muscles where the level of nuclear β -gal in muscle fibres nuclei is not significantly altered (increased) in *lppG4 > Pglym78-RNAi*. The filamentous signal represents background from sticky parts of the cuticle and should not be considered. We decided to leave these results out of the manuscript.

Figure for referees not shown.

ii) We have generated fly lines carrying UAS inducible over-expression of a constitutive form of *egr*, and of *ImpL2*. However, as reported previously in Sriskanthadevan-Pirahas et al. 2022 Cell Reports 39 page 6, the co-overexpression of *egr* and *ImpL2* using fat body drivers (*lpp-Gal4* here, or *R4-Gal4* previously) lead to strong early developmental defects, even at low temperatures, preventing us to assess whether this combination was sufficient to induce muscle disorganisation.

Citation of the study by Sriskanthadevan-Pirahas et al.: "However, expression of *Eiger* and *ImpL2* induced a strong impairment of growth and development, making it difficult to examine any effects of TFAM-regulated body growth."

Here is attached the image of 15d old larvae and 20d old pupae (for the animals that reach pupal stages), of overexpressing *ImpL2*, showing strong systemic effects blunting overall growth.

Figure for referees not shown.

5) The second paragraph in the discussion (page 11) starting with "Egr/TNF-alpha" appears too speculative. *Egr* mRNA levels are reduced. Additionally, it is not shown if TNF signaling is activated in the fat body or body wall muscles.

We thank the reviewer for this comment which stimulated us to perform additional experiments. In order to strengthen the case for *egr*/TNF- α signalling, we show now that circulating levels of *egr* (cleaved *egr*) were elevated in *Pglym78-RNAi* animals. This was shown by western-blot analyses of whole protein extracts from the haemolymph, using an unpublished anti-*egr* antibody kindly provided by Professor Konrad Basler, and normalised using *ApoLpp* (Figure 5B). The increase in active *egr* is an increase in concentration, since same volumes of haemolymph were blotted. Our interpretation is that while *egr* RNA levels appear decreased, *Tace* levels are increased which might account for the increase in *egr* processing. Interestingly, *REPTOR-RNAi* suppressed the increase in circulating *egr*, even though it did not have any impact on the changes in *egr* and *Tace* expressions, suggesting that REPTOR controls *egr* processing by other mechanisms, for instance by regulating *Tace* activity. With this report of increased circulating *egr* in the haemolymph, together with the functional requirement of *egr* and *Tace* for muscle defects, support more clearly the role for *egr*.

But it is true that we cannot show that *egr* will act directly on muscles. This is clearly stated in the discussion. While we could not monitor any significant changes to JNK activity in muscles, there was a strong activation of JNK activity reporters in *Pglym78-RNAi* fat body cells (see picture below).

Figure for referees not shown.

We thus now provide evidence of an increase in *egr* secretion, and a genetic requirement of *egr* and *Tace* in fat body cells for muscle disorganisation, even though, we cannot conclude that the *egr* produced by the fat body acts directly on muscles.

6) In the third paragraph of the discussion (page 11), "Alternatively, ageing". It is not clear how larval muscle degeneration phenotype is compared to muscle aging, given that their physiological contexts are completely different. Those sentences can be removed or should provide additional arguments.

We have now removed these two sentences from the Discussion

7) On page 12, "while the exact..... secreted by the fat body". No data is shown to support that REPTOR controls what nutrients are stored or secreted by the fat body. The sentence should be deleted or appropriately revised.

We have removed this sentence from the Discussion

8) It is not clearly described how glycogen and trehalose are measured and how they are normalized. Are they normalized to protein levels (in hemolymph)? Is concentration measured or the total amount?

We have extended the description in the Materials and Methods section, and included additional metabolite measurements. When metabolites were measured on dissected fat bodies (TAG, ATP, pyruvate), we normalised by total nucleic acid content (as a proxy to cell number). whole larvae, we normalised by the number. As stated earlier, normalisation for severely atrophied tissues, where almost everything is affected, represents a challenge.

But we also performed some measurements on whole larvae, normalising by the number of animals (total larvae size was relatively stable between genetic conditions).

More specifically, for Glycogen measurements shown in Fig. EV1 (and also for TAG), we made measurements on whole larvae, normalising by the number of animals. Indeed, we reasoned that the fat body represents the main organ storing glycogen (and TAGs), and that changes in fat body storing ability would be reflected in whole animal measurements. For Glycogen, we thus show here a reflection of the total amount per animal.

Trehalose measurements were performed on the similar amount of haemolymph (5µl), providing direct estimation of the circulating concentration when compared to a standard curve. Of note, on average all genotypes had similar volume of haemolymph per animal.

9) Did the authors test if RNAi lines used in the study efficiently knockdown the target mRNAs?

This is similar to comment #1 from Referee #1

We have included data on the RNAi knock-down of the different lines used in the Fig. EV1A. Of note, all of the glycolysis enzymes RNAi used were of similar strength with a knock-down efficiency greater than 80%. This has now been added in the text on page 5: "Importantly, the knock-down efficiencies of the different RNAi used were very similar and reached more than 80% for *Pgk*, *Pglym78*, and *Eno*, suggesting that the differences observed between *Pgk* and *Pglym78* were not caused by differences in RNA knock-down (Fig. EV1A)".

For the functional tests, the RNAi efficiencies were between 50% and 95%. Nevertheless we observed modifications of the phenotypes for *REPTOR*, *egr*, *Tace*, and *ImpL2* showing that even though partial, the knock down were sufficient to modify the *Pglm78-RNAi* phenotypes. For the Upd ligands, we removed the Upd1 and Upd2 results. Indeed, we felt this was providing too much negative results, and Upd1 and Upd2 have not been linked to muscle atrophy. In our muscle disorganisation modifier candidate approach we thus focus the text and results on Upd3 which has been linked to muscle atrophy in adult cachexia models. The knock-down of *upd3*, even though relatively strong (70%), was not able to restore “normal” levels similar to those found in a wild-type fat body, and we cannot exclude that the lack of modifications observed after knocking down *upd3* could be explained by its residual extra expression.

10) Based on the description in the method section, the data presented in Figure 4A show fold changes. How is the fold change calculated? If fold changes are shown, the y-axis label needs to be revised. It could be just an error, but the values for *lpp>wRNAi* are not 1.0. Is it because the values are normalized in a peculiar way? Would it be more appropriate to have the values for *lpp>wRNAi* normalized to be 1.0?

Thank you for pointing this out. This stemmed from the way we normalised the controls to allow to measure the standard deviation between the different replicates of the controls. We have now corrected this fold normalisation where controls are at 1.0

11) The study provides an interesting possibility of using hemolymph alpha-actinin as a molecular marker for muscle degeneration. It will make the finding very exciting to those who study cachexia using *Drosophila* tumor models if the authors show that alpha-actinin can be also detected in the hemolymph of larvae or adult flies bearing tumors inducing muscle degeneration.

Thank you for this supportive comment, but we feel that this falls beyond the scope of this study, looking at completely different models and physiological conditions.

Referee #3:

In this study, Rodriguez-Vazquez and colleagues investigate an inter-tissue communication mechanism between the fat body and muscle in *Drosophila* larvae. The authors utilize in vivo *Drosophila* genetic models, attenuating key glycolytic enzymes in the fat body, to induce fat body atrophy, which further leads to muscle disorganization (wasting) through a potential systemic mechanism - implicating TOR signaling in the fat body, as well as Eiger (TNF) and ImpL2, as part of a communication signaling network between fat body and muscle in *Drosophila* larvae.

Overall, the manuscript attempts to address an interesting question related to growth coordination during development, potentially linking key metabolic cycles in nutrient storage tissues to proper growth of distal/peripheral tissues through distinct signaling mechanism.

However, I have some major conceptual and experimental concerns with the manuscript, which include a general lack of clarity related to the biological relevance of the findings (associated with both the genetic model of glycolysis inhibition and the tissue communication axis) and a lack of sufficient experimental justification for many of the conclusions related to mechanism. Below are some major concerns:

1. The authors induce fat body atrophy through inhibiting key glycolytic enzymes, which drives autonomous lipid metabolic dysfunction and at least one sign of systemic muscle wasting. These genetic models lead to either very mild or severe developmental phenotypes. However, it's unclear if glycolysis is an important feature of this inter-organ communication, or just an artificial model to severely limit fat body function. This diminishes the overall biological relevance of the larger phenotypic findings and mechanisms. Just as a few examples:

While the authors supplement pyruvate in the fly food and show a rescue of fat body atrophy in *LppGal4>Pglym78* or *Eno RNAi* larvae, they never check if pyruvate supplementation impacts developmental phenotypes, muscle wasting, or changes in gene expression (used mainly to imply mechanism).

Pyruvate feeding rescued the atrophy of fat body cells, as shown by cell size rescue and lipid droplets content. But pyruvate feeding also rescued muscle morphology, at least in part. This is now shown in Figure EV3 and discussed in the text.

The authors never measure pyruvate or other metabolites in their various genetic models to explore how glycolysis (or other impacted metabolic cycles) are impacted - maybe important since carbohydrate storage (and circulation) was not impacted?. Etc.

See also comments #2 & 3 from Referee #1

In this study, we focussed on glycolysis enzymes. We now show steady state measurements for triglycerides, pyruvate, and ATP. We did show in the original version, lipid droplets, glycogen, and circulating trehalose. We would like to stress that the primary focus of this study is to explore the potential distant effects of metabolic alteration (here glycolysis) in the fat body on the physiology of the larvae and identify the inter-organ exchanges. It was not to describe the metabolic rewiring occurring inside the adipocyte cells. Thus, while interesting, detailed metabolic analyses studying how the different metabolic pathways are affected is out of the scope of the current study and would represent a project on its own.

We measured pyruvate levels in the atrophied *Pglym78-RNAi* fat body, and show that steady state levels normalised to total nucleic acid content actually increase following glycolysis impairment. We interpret this result as a sign that upon *Pglym78-RNAi*, cells rewired their metabolism and stopped using pyruvate to generate energy. Pyruvate, the final product of glycolysis is no longer used and accumulates even if produced at a reduced rate due to the *Pglym78-RNAi*. Indeed, when we supplemented the food with 1% pyruvate, the steady state levels of pyruvate go back to normal, indicative that pyruvate is now again used as a source of Ac-CoA and energy. This is now shown in Fig. EV1.

Since TAG levels were decreased and that lipid droplets shrank (BODIPY), the metabolic rewiring is likely a shift to β -oxidation of lipids as source of Ac-CoA which can then fuel the TCA cycle. This shift from glycolysis to β -oxidation as source of energy is now discussed in the results and in the discussion on pages 5 and 12.

2. Related to the over-arching mechanisms and the proposed model in Figure 7: in general, there is an over-reliance on genetic interactions and a lack of supplemental experimental analysis that would support the mechanistic conclusions provided in the manuscript. For example, while an increase in REPTOR activity was never explicitly shown, the authors conclude that 'these results show that upon glycolysis shutdown, adipocyte atrophy is mediated, at least in part, by increased REPTOR activity, leading to distant muscle disorganization'.

See comment #3 from referee #2

We agree with the referee, that REPTOR activity was not measured in the original version of the manuscript. Even though the target genes of REPTOR in larval fat body cells have not been formally identified, we have used two commonly reported REPTOR transcriptional targets as REPTOR activity reporters: *4E-BP/thor* and *unk*. *4E-BP/thor* is also controlled by other pathways including Insulin/FOXO signalling, but *unk* represents a relatively specific target gene of REPTOR.

We show here that mTOR pathway activity is lower in *Pglym78-RNAi* fat body cells. However, *4E-BP* and *unk* transcripts levels were down, suggesting that REPTOR activity is down-regulated in *Pglym78-RNAi* cells. We propose that this might be due to lower levels of REPTOR-BP, whose transcripts levels are low in *Pglym78-RNAi* cells. We propose in the discussion that this low level of REPTOR/REPTOR-BP might represent an adaptation of cells to prevent excessive catabolism in conditions where cells have already almost used up their lipid reserves. These results have been added in Fig 4C.

However, while REPTOR activity was diminished in *Pglym78-RNAi* cells, it was still active since lowering REPTOR levels by RNAi further diminished *4E-BP* and *unk* expression levels (Fig. 4D). Removing REPTOR also efficiently suppressed fat body atrophy suggesting that either REPTOR activity was critical at earlier stages for atrophy, or that the remaining REPTOR activity participates in the fat body atrophy.

We have included all these new data, suggesting that the link between mTOR and REPTOR might be more complicated in the context of *Pglym78-RNAi*

What is the connection between glycolytic inhibition and the TOR pathway?, is it direct or indirect?

In *Pglym78-RNAi* animals, we show that the expressions of several genes coding for important TORC1 regulators or core components (Tor, raptor, Rheb) are down-regulated. This is proposed as potential link between the glycolysis inhibition and TORC1 inhibition in the Discussion (pages 14-15). In the original Discussion we also referred to different studies showing a link between the energy sensor AMPK and mTOR regulation. However, during the revision process, we now show that ATP levels were increased in *Pglym78* knock-down (Fig. EV1). Furthermore, pAMPK levels were also down in *Pglym78-RNAi* animals (Fig. 4B). Even though we could not normalise by the total AMPK, this result could indicate that AMPK activity was not increased after glycolysis inhibition in fat body cells. We have thus updated this part from the Discussion. The mTOR pathway represents actually an integrator of many different signals to fine tune metabolic activity in response to external and internal stimuli. The field of research into mTOR regulation is extremely active describing regularly novel intricate cross-regulations, such as AMPK, P53, aa transports, aa metabolism, Insulin pathway... that can be sensitive to glucose levels... This represents an independent project, out of the scope of the current study, which would probably be better addressed in a simpler system such as cultured cells.

, would this be true of any genetic model which induced fat body atrophy, and not specific to glycolysis? These same questions can be asked of the mechanistic link between glycolytic-induced fat body atrophy and changes in Eiger or ImpL2. Thus, it is impossible to determine the directness or indirectness of some of these potential mechanisms, whether it be the link between glycolysis and TOR/REPTOR activity, the link between glycolysis/TOR and Eiger or ImpL2 secretion/function, etc. In other the words, the arrows in the proposed model, as suggested, are inconclusive as it relates to the interconnectedness of these mechanism(s), making the study very correlative and potentially (without further experimental analysis) a bit artificial.

The comment of the referee is not very clear, but we think the referee is here questioning the specificity of the effects observed. This argument appears somehow unfair.

First, a cell, a tissue, a biological system, will respond to perturbations in an integrated manner and according to its competence. The integrated response of systems can be seen as the interconnectedness of processes the referee is referring to. Whether different alterations can cause changes in mTOR activity or *egr/ImpL2* function in fat body cells is possible, but we argue that this does not undermine the need to identify and study the different signals that can lead to these responses.

Second, if we understood the points of the referee, the lack of specificity of our study is also in part due to a lack of more intricate molecular mechanistic description. We provide molecular evidence that

activated *egr* protein accumulates in the haemolymph, and that *egr* and *Tace* are required, making a continuum between genetic and molecular experiments. We provide now a link between REPTOR and *egr* secreted levels linking the part between REPTOR and adipokines. We also provide molecular evidence that Insulin signalling is altered in Muscles, and genetic evidence that *ImpL2*, an insulin trap, is required in the fat body. We acknowledge in the Discussion that we cannot conclude that the effects of the molecules secreted by the fat body are direct or indirect, or how the glycolysis impinge on mTOR activity, in large part due to limitation of the whole animal system we use. Even though the systems have limitations, whole animal systems ensure that the responses and effects observed are biologically relevant and cannot be called artificial.

I also find it odd that an entire main Figure was used to highlight that *Upd3* is likely not inducing muscle wasting (negative result), when the next Figure does highlight a potential secreted factor that could be driving this inter-organ communication.

Following the referee's comment we have moved all aspects regarding *upd3* as Fig. EV4 & EV5. We have removed the results on *upd1* and *upd2*. Indeed, we felt this was providing too much negative results, and *upd1* and *upd2* have not been linked to muscle atrophy. In our muscle disorganisation modifier candidate approach we thus focus the text and results on *upd3* which has been linked to muscle atrophy in adult cachexia models.

3. In general, more data (controls and otherwise) needs to be provided to justify some of the findings. Just as a few examples: The authors should include additional measurements for muscle disorganization/wasting (especially linked to function) and specific mechanisms underlying muscle wasting should be discussed.

See comments #1 & 4 from referee #2

We provide the quantification of body wall muscle disorganisation phenotype in the figures and legends. Quantification methods have been added in the Material and Methods section and are based on previously published method in Newton et al. Nat Commun 2020 (doi:10.1038/s41467-020-18502-9). Briefly, muscle disruption was categorised into: none, minor, medium, strong disruption. Statistics on frequency changes compared to the expected frequencies in *white-RNAi* or *Pglym78-RNAi* controls were assessed using the chi squared statistical test. We report that the frequency of medium and strongly affected muscles increases in *Pglym78* or *Eno RNAi* compared to *white-RNAi* controls, and that this increase is suppressed when deleting *egr*, *Tace*, or *ImpL2*.

Following the comments of referee#2 we have explored whether JNK pathway or Caspases were activated in the muscles, but we did not observe any significant differences (see response there). It appears that muscle disorganisation does not rely on Caspase-mediated apoptosis (which is actually consistent with our ability to recover what appeared full length muscular protein in the haemolymph).

Why is there such a massive difference in developmental phenotypes when attenuating *Eno* vs *Pglym78*, and are these potential different models of metabolic deficiency and thus influencing different types of mechanisms? Etc.

We followed the level of knock-down of *Pglym78* and *Eno* RNAi by qPCR. They both induce robust knock down of around 80%, with *Eno* RNAi being slightly stronger (Fig. EV1A). The differences in phenotypes regarding developmental delay or fat body atrophy might reflect these small differences in RNAi efficiency. It should be noted that we only monitored RNA levels, and it remains possible that the differences in term of strength of phenotypes originate from differences in the relative abundancies of the two genes, or that protein stabilities are different, or that their dosage requirements are different.

Eno-RNAi gives a more pronounced phenotype both for developmental delay and for adipocyte cell atrophy. Thus, in all likelihood *Pglym78-RNAi* and *Eno-RNAi* represent two degrees of the same phenotypic alteration: fat body atrophy and its consequences: developmental delay and muscle disorganisation.

It is also possible that both enzymes have specific functions affecting different processes besides glycolysis (and the glycolysis-blockade consequences we report here in this study), but this falls out of the scope of this study.

Dear Dr. Djiane,

Thank you for submitting your revised manuscript. It has now been seen by all of the original referees.

As you can see, the referees find that the study is significantly improved during revision and recommend publication. However, I need you to address the points below before I can accept the manuscript.

- Please rename the Competing Interests section as "Disclosure Statement and Competing Interests".
- Please remove the Author Contributions section from the manuscript.
- As per our format requirements, in the reference list, citations should be listed in alphabetical order and then chronologically, with the authors' surnames and initials inverted; where there are more than 10 authors on a paper, 10 will be listed, followed by 'et al.'. Please see <https://www.embopress.org/page/journal/14693178/authorguide#referencesformat>
- DOIs need to be removed, which are only needed for preprints and datasets that have not been published yet.
- Funding information should be a part of the Acknowledgements section. Therefore, the 'Funding' title needs to be removed.
- We note the following regarding figure callouts: Figure 4J is currently not called out in the text. 'Supplemental Fig. S3A' callout needs to be corrected as Fig. EV3A.
- The following corrections should be done on section titles: Summary should be Abstract. Materials and Methods should be Methods.
- Our production/data editors have asked you to clarify several points in the figure legends:
 - o Please note that the legend for figure 4j is missing in the manuscript. This needs to be rectified.
 - o Please note that the exact p values are not provided in the legends of figures 1c-e; 3c; 4a, c-d, g-h; 5e; 6a-b, e-f; EV 1a, c, f-h; EV 3e; EV 4a, e; EV 5b.
 - o Please note that in figures 1c-d; 4a; 5a; there is a mismatch between the annotated p values in the figure legend and the annotated p values in the figure file that should be corrected.
 - o Please note that the box plots need to be defined in terms of minima, maxima, centre, bounds of box and whiskers, and percentile in the legends of figures 1c-d; 4g-h; 5e; 6e-f; EV 1c; EV 4e; EV 5b, f.
 - o Please note that information related to n is missing in the legends of figures 1c-d; 2b; 3c; 4a, c-d, g-h; 5e; 6e; EV 1c; EV 2b; EV 4e; EV 5b, f.
 - o Please note that the error bars are not defined in the legends of figures 1e; 2b; 3c
 - o Please note that the scale bar is missing for figures 1b'; 4e'; 5c'; 6c'; EV 4c'; EV 5a', d'.
 - o Please note that the scale bar needs to be defined for figures EV 1d; EV 3a.
 - o Please note that in the legend for figure EV 3d, red arrowhead is defined in the legend, however the same is not marked in the figure. This needs to be rectified.
- Papers published in EMBO Reports include a 'synopsis' and 'bullet points' to further enhance discoverability. Both are displayed on the html version of the paper and are freely accessible to all readers. The synopsis includes a short standfirst summarizing the study in 1 or 2 sentences (max 35 words) that summarize the paper and are provided by the authors and streamlined by the handling editor. I would therefore ask you to include your synopsis blurb and 3-5 bullet points listing the key experimental findings.
- In addition, please provide an image for the synopsis. This image should provide a rapid overview of the question addressed in the study but still needs to be kept fairly modest since the image size cannot exceed 550 (width) x 300-600 (height) pixels.

Thank you again for giving us to consider your manuscript for EMBO Reports, I look forward to your minor revision.

Kind regards,

Deniz Senyilmaz Tiebe

--

Deniz Senyilmaz Tiebe, PhD
Scientific Editor
EMBO Reports

Referee #1:

The authors have addressed my concerns.

Referee #2:

The authors have appropriately addressed my comments. The revised manuscript should be suitable for publication.

Referee #3:

The authors have added new data and text to this revision which has definitely improved the overall quality of the manuscript and interpretation of data. I still contend that depth of mechanistic discovery is lacking, even with acknowledging that the authors have argued that these represent complex biological processes that might go beyond the scope of this manuscript. However, based on the perspective of other reviewers, I agree that this manuscript provides interesting new findings relevant to the expanding field of insect developmental metabolism.

Ref: EMBOR-2023-58393-T

Dear Dr Senyilmaz Tiebe,

We are extremely happy that the reviewers recommended that our manuscript should be published in Embo Reports. Please find below our answers to the comments. They are highlighted in red in the resubmitted manuscript and figures.

We thus resubmit the Main Text, and Figures 1, 4, 5, 6, EV1, EV3, EV4, EV5 with the corrections requested.

Best Regards,

Alexandre Djiane

#####

Comments:

As you can see, the referees find that the study is significantly improved during revision and recommend publication. However, I need you to address the points below before I can accept the manuscript.

- Please rename the Competing Interests section as "Disclosure Statement and Competing Interests".

This has been changed

- Please remove the Author Contributions section from the manuscript.

This section has been removed

- As per our format requirements, in the reference list, citations should be listed in alphabetical order and then chronologically, with the authors' surnames and initials inverted; where there are more than 10 authors on a paper, 10 will be listed, followed by 'et al.'. Please see

<https://www.embopress.org/page/journal/14693178/authorguide#referencesformat>
DOIs need to be removed, which are only needed for preprints and datasets that have not been published yet.

The references have now been formatted accordingly

- Funding information should be a part of the Acknowledgements section. Therefore, the 'Funding' title needs to be removed.

The Funding section has been integrated into the Acknowledgements

- We note the following regarding figure callouts: Figure 4J is currently not called out in the text. 'Supplemental Fig. S3A' callout needs to be corrected as Fig. EV3A.

We are sorry for that mistake. The Figure 4J was mis-called as Figure 4H. This has now been fixed on page 8.

The Supplemental Fig. S3A that remained on page 18 in the Methods section has been corrected as Fig. EV3A.

- The following corrections should be done on section titles: Summary should be Abstract. Materials and Methods should be Methods.

This has now been changed

- Our production/data editors have asked you to clarify several points in the figure legends:

o Please note that the legend for figure 4j is missing in the manuscript. This needs to be rectified.

The legend has been added

o Please note that the exact p values are not provided in the legends of figures 1c-e; 3c; 4a, c-d, g-h; 5e; 6a-b, e-f; EV 1a, c, f-h; EV 3e; EV 4a, e; EV 5b.

We provide the exact p-value in the legend, when it is superior to 0.0001.

1c (**** <0.0001, ***0.0006, **0.01), 1d (****<0.0001, ***0.001), 1e(*0.0286)

3c (****<0.0001, ***0.001)

4a (****<0.0001, ***0.0001, **0.0026), 4c(****<0.0001, ***0.0001), 4d(***0.001, *0.0240),

4g(****<0.0001), 4h(****<0.0001), 4j(***0.001)

5a(****<0.0001, **0.0010), e(****<0.0001), 5f(****<0.0001, **0.0028)

6a(****<0.0001), 6b(****<0.0001, **0.0010), 6e(****<0.0001, ***0.0010), 6f(**0.0034)

EV 1a(****<0.0001), c(**0.0043), f(*0.0260), g(**0.0079), h(*0.0374)

EV 3e(***0.0002, *0.0156)

EV 4a(****<0.0001), e(****<0.0001, ***0.0003)

EV 5b(****<0.0001)

o Please note that in figures 1c-d; 4a; 5a; there is a mismatch between the annotated p values in the figure legend and the annotated p values in the figure file that should be corrected.

This has been corrected.

o Please note that the box plots need to be defined in terms of minima, maxima, centre, bounds of box and whiskers, and percentile in the legends of figures 1c-d; 4g-h; 5e; 6e-f; EV 1c; EV 4e; EV 5b, f.

This has now been added in the legends. Box plots were as follow: whiskers represent the maxima and minima experimental points, the boxes represent the 25% and 75% percentiles, and the centre line is the median

o Please note that information related to n is missing in the legends of figures 1c-d; 2b; 3c; 4a, c-d, g-h; 5e; 6e; EV 1c; EV 2b; EV 4e; EV 5b, f.

1c-d (n=6-10)

2b (brains n=15/13, wing discs n=26/27)

3c (n>11)

4a (n=3), c (n=3), d (n=3), g-h (n=10), j (n>10)

5a (n=3), e (n=6-9), f (n>11)

6a (n=3), b (n=3), e (n=10-14), f (n=10)

EV 1c (n=7-11)

EV 2b (brains n=13/10, wing discs n=27/24)

EV 4e (n=7-11), f (n>10)

EV 5b (n=6-11), f (n=8)

o Please note that the error bars are not defined in the legends of figures 1e; 2b; 3c

In 1e and 2b, they represent standard error to the mean (sem). This is now in the legend.

We do not understand the point for Fig 3c, as there is no error bar in this panel.

o Please note that the scale bar is missing for figures 1b'; 4e'; 5c'; 6c'; EV 4c'; EV 5a', d'.

Most of the scale bars missing correspond to the primed number panels. They represent the obvious split channels from the main panels, and are obviously the same magnification. Anyway, we added the requested scale bars. The legends did not need to be amended.

o Please note that the scale bar needs to be defined for figures EV 1d; EV 3a.

These scale bars have been now added in the figures and the legends have been modified accordingly.

EV 1d : 200µm

EV 3a : 200µm

o Please note that in the legend for figure EV 3d, red arrowhead is defined in the legend, however the same is not marked in the figure. This needs to be rectified.

This has been rectified.

- Papers published in EMBO Reports include a 'synopsis' and 'bullet points' to further enhance discoverability. Both are displayed on the html version of the paper and are freely accessible to all readers. The synopsis includes a short standfirst summarizing the study in 1 or 2 sentences (max 35

words) that summarize the paper and are provided by the authors and streamlined by the handling editor. I would therefore ask you to include your synopsis blurb and 3-5 bullet points listing the key experimental findings.

Glycolysis impairment in the fat body of developing *Drosophila* larvae leads to distant body-wall muscle disorganization. These muscle defects are mediated by the adipose tissue secretion of TNF- α /egr and the insulin signalling extracellular inhibitor ImpL2.

- Fat body (FB) knock-down (KD) of the glycolysis enzyme Pglym78 or Eno results in lipid storage shrinkage and adipocytes atrophy
- Pglym78 FB KD leads to developmental delay and distant muscle disorganization
- REPTOR activity in FB cells is required for adipocytes atrophy and for distant muscle disorganization
- Muscle defects are mediated by the secretion by FB cells of TNF- α /egr and ImpL2
- REPTOR controls egr circulating levels in the hemolymph

• In addition, please provide an image for the synopsis. This image should provide a rapid overview of the question addressed in the study but still needs to be kept fairly modest since the image size cannot exceed 550 (width) x 300-600 (height) pixels.

Please find attached the image requested. It also uploaded on the submission website.

Dr. Alexandre Djiane
Inserm
Institut de Recherche en Cancerologie de Montpellier
208 rue des apothicaires
Montpellier 34298
France

Dear Dr. Djiane,

Thank you for submitting your revised manuscript. I have now looked at everything and all is fine. Therefore, I am very pleased to accept your manuscript for publication in EMBO Reports.

Congratulations on a nice work!

Kind regards,

Deniz Senyilmaz Tiebe

--

Deniz Senyilmaz Tiebe, PhD
Senior Scientific Editor
EMBO Reports

--
